# Estimating the Effects of Continuous-valued Interventions using Generative Adversarial Networks

**Ioana Bica**[*]
Department of Engineering Science
University of Oxford, Oxford, UK
The Alan Turing Institute, London, UK
ioana.bica@eng.ox.ac.uk

**James Jordon**[*]
Department of Engineering Science
University of Oxford, Oxford, UK
james.jordon@wolfson.ox.ac.uk

**Mihaela van der Schaar**
University of Cambridge, Cambridge, UK
University of California, Los Angeles, USA
The Alan Turing Institute, London, UK
mv472@cam.ac.uk

## Abstract

While much attention has been given to the problem of estimating the effect of discrete interventions from observational data, relatively little work has been done in the setting of continuous-valued interventions, such as treatments associated with a dosage parameter. In this paper, we tackle this problem by building on a modification of the generative adversarial networks (GANs) framework. Our model, SCIGAN, is flexible and capable of simultaneously estimating counterfactual outcomes for several different continuous interventions. The key idea is to use a significantly modified GAN model to learn to generate counterfactual outcomes, which can then be used to learn an inference model, using standard supervised methods, capable of estimating these counterfactuals for a new sample. To address the challenges presented by shifting to continuous interventions, we propose a novel architecture for our discriminator - we build a hierarchical discriminator that leverages the structure of the continuous intervention setting. Moreover, we provide theoretical results to support our use of the GAN framework and of the hierarchical discriminator. In the experiments section, we introduce a new semi-synthetic data simulation for use in the continuous intervention setting and demonstrate improvements over the existing benchmark models.

## 1 Introduction

Estimating the personalised effects of interventions is crucial for decision making in many domains such as medicine, education, public policy and advertising. Such domains have a wealth of observational data available. Most of the methods developed in the causal inference literature focus on learning the counterfactual outcomes of discrete interventions, such as binary or categorical treatments[2] [1–9]. Unfortunately, in many cases, deciding how to intervene involves not only deciding which intervention to make (e.g. whether to treat cancer with radiotherapy, chemotherapy or surgery) but also deciding on the value of some continuous parameter associated with intervening (e.g. the dosage of radiotherapy to be administered). In medicine there are many examples of treatments

---

[*]Equal contribution.
[2]For ease of exposition, we will sometimes refer to interventions as treatments and to the associated continuous parameter as the dosage throughout the paper.

that are associated with a continuous dosage parameter (such as vasopressors [10]). In the medical setting, using a high dosage for a treatment can lead to toxic effects while using a low dosage can result in no effect on the patient outcome [11, 12]. In other domains, there are many examples of continuous interventions, such as the duration of an education or job training program, the frequency of an exposure or the price used in an advert. Naturally, being able to estimate the effect of these continuous interventions will aid in the decision making process.

Learning from observational data already presents significant challenges when there is only a single intervention (and thus the decision is binary - whether to intervene or not). As explained in [13], in an observational dataset, only the factual outcome is present - the "counterfactual" outcomes are not observed. This problem is exacerbated in the setting of continuous interventions where the number of counterfactuals is no longer even finite. Moreover, the decision to intervene is non-random and instead is assigned according to the features associated with each sample. Due to the continuous nature of the interventions, adjusting for selection bias is significantly more complex than for binary (or even multiple) interventions. Thus, standard methods for adjusting for selection bias for discrete treatments cannot be easily extended to handle bias in the continuous setting.

We propose SCIGAN (eStimating the effects of Continuous Interventions using GANs). We build on the GAN framework of [14] to learn the distribution of the unobserved counterfactuals. GANs have already been used in GANITE [6] to generate the unobserved counterfactual outcomes for discrete interventions. The intuition is that if a counterfactual generator and discriminator are trained adversarially, then the generator can fool the discriminator (i.e. the discriminator will not be able to correctly identify the factual outcome) by generating counterfactuals according to their true distribution. Unfortunately, no theoretical work was provided in [6] to back up this

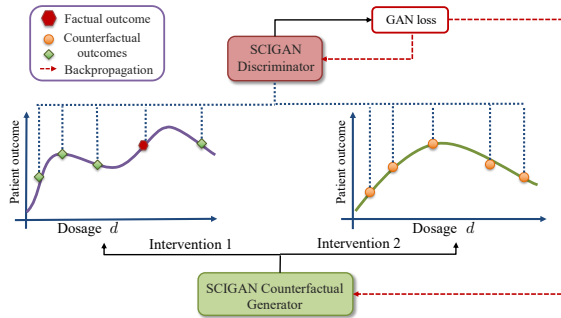

Figure 1: Overview of GAN framework used as part of SCIGAN for learning the distribution of the counterfactual outcomes.

intuition. A key contribution of this paper is to provide theoretical results that justify using the GAN framework to learn to generate counterfactual outcomes; these results also apply to GANITE.

GANITE itself presents a significant modification to the original GAN framework - rather than the discriminator discriminating between entirely real or entirely fake samples, the discriminator is attempting to identify the real component from a vector containing the real (factual) outcome from the dataset and the fake (counterfactual) outcomes generated by the generator. SCIGAN inherits this key difference from a standard GAN. However, beyond our theoretical contribution, we propose significant changes to the generator and discriminator in order to tackle the more complex problem of estimating outcomes of continuous interventions.

We define a discriminator that acts on a finite set of points from each generated response curve (rather than on entire curves), as shown in Fig. 1. We draw on ideas from [15] to ensure that our discriminator acts as a function of a set, rather than of a vector. Moreover, we focus on the setting of interventions *each* with an associated continuous parameter. In this setting, we propose a *hierarchical* discriminator which breaks down the job of the discriminator into determining the factual intervention and determining the factual parameter using separate networks. We show in the experiments section that this approach significantly improves performance and is more stable than using a single network discriminator. We also model the generator as a multi-task deep network capable of taking a continuous parameter as an input; this gives us the flexibility to learn heterogeneous response curves for the different interventions.

Our contributions in this paper are 4-fold: (1) we propose SCIGAN, a significantly modified GAN framework, capable of estimating outcomes for continuous and many-level-discrete interventions, (2) we provide theoretical justification for both the use of a GAN framework and a hierarchical discriminator, (3) we propose novel architectures for each of our networks, (4) we propose a new semi-synthetic data simulation for use in the continuous intervention setting. We show, using semi-synthetic experiments, that our model outperforms existing benchmarks.

## 2 Related work

Methods for estimating the outcomes of treatments with a continuous dosage from observational data make use of the generalized propensity score (GPS) [16–18] or build on top of balancing methods for multiple treatments. [19] developed a neural network based method to estimate counterfactuals for multiple treatments and continuous dosages. The proposed Dose Response networks (DRNets) in [19] consist of a three level architecture with shared layers for all treatments, multi-task layers for each treatment and additional multi-task layers for dosage sub-intervals. Specifically, for each treatment $w$, the dosage interval $[a_w, b_w]$ is subdivided into $E$ *equally* sized sub-intervals and a multi-task head is added for each sub-interval. This is an extension of the architecture in [20]. However, the main advantage of using multi-task heads for dosage intervals would be the added flexibility in the model to learn potentially very different functions over different regions of the dosage interval. DRNets do not determine these intervals dynamically and thus much of this flexbility is lost. Our approach (using GANs to generate counterfactuals) fundamentally differs from DRNets (supervised learning with bias-adjustment) and we demonstrate experimentally that SCIGAN outperforms GPS and DRNets.

For a discussion of works that address treatment-response estimation without a dosage parameter, see Appendix A. Note that for such methods we cannot treat the dosage as an input due to the bias associated with its assignment. In Appendix A, we also describe the relationships between our work (causal inference for continuous interventions) and policy optimization with continuous treatments.

## 3 Problem formulation

We consider receiving observations of the form $(\mathbf{x}^i, t_f^i, y_f^i)$ for $i = 1, ..., N$, where, for each $i$, these are independent realizations of the random variables $(\mathbf{X}, T_f, Y_f)$. We refer to $\mathbf{X}$ as the feature vector lying in some feature space $\mathcal{X}$, containing pre-treatment covariates (such as age, weight and lab test results). The treatment random variable, $T_f$, is in fact a pair of values $T_f = (W_f, D_f)$ where $W_f \in \mathcal{W}$ corresponds to the *type* of treatment being administered (e.g. chemotherapy or radiotherapy) which lies in the discrete space of $k$ treatments, $\mathcal{W} = \{w_1, ..., w_k\}$, and $D_f$ corresponds to the *dosage* of the treatment (e.g. number of cycles, amount of chemotherapy, intensity of radiotherapy), which, for a given treatment $w$ lies in the corresponding treatment's dosage space, $\mathcal{D}_w$ (e.g. the interval $[0, 1]$). We define the set of all treatment-dosage pairs to be $\mathcal{T} = \{(w, d) : w \in \mathcal{W}, d \in \mathcal{D}_w\}$.

Following Rubin's potential outcome framework [21], we assume that for all treatment-dosage pairs, $(w, d)$, there is a potential outcome $Y(w, d) \in \mathcal{Y}$ (e.g. 1-year survival probability). The *observed* outcome is then defined to be $Y_f = Y(W_f, D_f)$. We will refer to the unobserved (potential) outcomes as *counterfactuals*.

The goal is to derive *unbiased* estimates of the potential outcomes for a given set of input covariates:

$$\mu(t, \mathbf{x}) = \mathbb{E}[Y(t)|\mathbf{X} = \mathbf{x}] \tag{1}$$

for each $t \in \mathcal{T}$, $\mathbf{x} \in \mathcal{X}$. We refer to $\mu(\cdot)$ as the individualised dose-response function. A table summarising our notation is given in Appendix B. In order to ensure that this quantity is equal to $\mathbb{E}[Y|\mathbf{X} = \mathbf{x}, T = t]$ and that the dose-response function is identifiable from the observational data, we require the following two assumptions.

**Assumption 1.** *(Unconfoundedness) The treatment assignment, $T_f$, and potential outcomes, $Y(w, d)$, are conditionally independent given the covariates $\mathbf{X}$, i.e. $\{Y(w, d)|w \in \mathcal{W}, d \in \mathcal{D}_w\} \perp\!\!\!\perp T_f|\mathbf{X}$.*

**Assumption 2.** *(Overlap) $\forall \mathbf{x} \in \mathcal{X}$ such that $p(\mathbf{x}) > 0$, we have $1 > p(t|\mathbf{x}) > 0$ for each $t \in \mathcal{T}$.*

## 4 SCIGAN

We propose estimating $\mu$ by first training a generator to generate response curves for each sample *within* the training dataset. The learned generator can then be used to train an inference network using standard supervised methods. We build on the idea presented in [6], using a modified GAN framework to generate potential outcomes conditional on the observed features, treatment and factual outcome. Several changes must be made to both the generator and discriminator architectures and learning paradigms in order to produce a model capable of handling the dose-response setting.

### 4.1 Counterfactual Generator

Our generator, $\mathbf{G} : \mathcal{X} \times \mathcal{T} \times \mathcal{Y} \times \mathcal{Z} \to \mathcal{Y}^{\mathcal{T}}$ takes features, $\mathbf{x} \in \mathcal{X}$, factual outcome, $y_f \in \mathcal{Y}$, received treatment and dosage, $t_f = (w_f, d_f) \in \mathcal{T}$, and some noise, $\mathbf{z} \in \mathcal{Z}$ (typically multivariate uniform or Gaussian), as inputs. The output will be a dose-response curve for each treatment (as shown in Fig. 1), so that the output is a function from $\mathcal{T}$ to $\mathcal{Y}$, i.e. $\mathbf{G}(\mathbf{x}, t_f, y_f, \mathbf{z})(\cdot) : \mathcal{T} \to \mathcal{Y}$. We can then write

$$\hat{y}_{cf}(t) = \mathbf{G}(\mathbf{x}, t_f, y_f, \mathbf{z})(t) \tag{2}$$

as our generated counterfactual outcome for treatment-dosage pair $t$. We write $\hat{Y}_{cf}(t) = \mathbf{G}(\mathbf{X}, T_f, Y_f, \mathbf{Z})(t)$ (i.e. the random variable induced by $\mathbf{G}$).

While the job of the counterfactual generator is to generate outcomes for the treatment-dosage pairs which were *not* observed, [6] demonstrated that the performance of the counterfactual generator is improved by adding a supervised loss term that regularises its output for the factual treatment (in our case treatment-dosage pair). We define the supervised loss, $\mathcal{L}_S$, to be

$$\mathcal{L}_S(\mathbf{G}) = \mathbb{E}\left[(Y_f - \mathbf{G}(\mathbf{X}, T_f, Y_f, \mathbf{Z})(T_f))^2\right], \tag{3}$$

where the expectation is taken over $\mathbf{X}, T_f, Y_f$ and $\mathbf{Z}$.

### 4.2 Counterfactual Discriminator

As noted in Section 1, our discriminator will act on a random set of points from each of the generated dose-response curves. Similar to [6], we define a discriminator, $\mathbf{D}$, that will attempt to pick out the factual treatment-dosage pair from among the (random set of) generated ones. To handle the complexity arising from multiple treatments, we break down our discriminator into two distinct models: a treatment discriminator and a dosage discriminator.

Formally, let $n_w \in \mathbb{Z}^+$ be the number of dosage levels we will compare for treatment $w \in \mathcal{W}$[3]. For each $w \in \mathcal{W}$, let $\tilde{\mathcal{D}}_w = \{D_1^w, ..., D_{n_w}^w\}$ be a random subset[4] of $\mathcal{D}_w$ of size $n_w$, where for the factual treatment, $W_f$, $\tilde{\mathcal{D}}_{W_f}$ contains $n_{W_f} - 1$ random elements along with $D_f$. We define $\tilde{\mathbf{Y}}_w = (D_i^w, \tilde{Y}_i^w)_{i=1}^{n_w} \in (\mathcal{D}_w \times \mathcal{Y})^{n_w}$ to be the vector of dosage-outcome pairs for treatment $w$ where

$$\tilde{Y}_i^w = \begin{cases} Y_f \text{ if } W_f = w \text{ and } D_f = D_i^w \\ \hat{Y}_{cf}(w, D_i^w) \text{ else} \end{cases} \tag{4}$$

and $\tilde{\mathbf{Y}} = (\tilde{\mathbf{Y}}_w)_{w \in \mathcal{W}}$. We will write $d_j^w$, $\tilde{\mathbf{y}}_w$ and $\tilde{\mathbf{y}}$ to denote realisations of $D_j^w$, $\tilde{\mathbf{Y}}_w$ and $\tilde{\mathbf{Y}}$.

The treatment discriminator, $\mathbf{D}_{\mathcal{W}} : \mathbf{X} \times \prod_{w \in \mathcal{W}}(\mathcal{D}_w \times \mathcal{Y})^{n_w} \to [0, 1]^k$, takes the features, $\mathbf{x}$, and generated potential outcomes, $\tilde{\mathbf{y}}$, and outputs a probability for each treatment, $w_1, ..., w_k$. Writing $\mathbf{D}_{\mathcal{W}}^w$ to denote the output of $\mathbf{D}_{\mathcal{W}}$ corresponding to treatment $w$, we define the loss, $\mathcal{L}_{\mathcal{W}}$, to be

$$\mathcal{L}_{\mathcal{W}}(\mathbf{D}_{\mathcal{W}}; \mathbf{G}) = -\mathbb{E}\left[\sum_{w \in \mathcal{W}} \mathbb{I}_{\{W_f = w\}} \log \mathbf{D}_{\mathcal{W}}^w(\mathbf{X}, \tilde{\mathbf{Y}}) + \mathbb{I}_{\{W_f \neq w\}} \log(1 - \mathbf{D}_{\mathcal{W}}^w(\mathbf{X}, \tilde{\mathbf{Y}}))\right], \tag{5}$$

where the expectation is taken over $\mathbf{X}, W_f, D_f, \tilde{\mathbf{Y}}$ and $\{\tilde{\mathcal{D}}_w\}_{w \in \mathcal{W}}$.

Then, for each $w \in \mathcal{W}$, the dosage discriminator, $\mathbf{D}_w : \mathcal{X} \times (\mathcal{D}_w \times \mathcal{Y})^{n_w} \to [0, 1]^{n_w}$, is a map that takes the features, $\mathbf{x}$, and generated potential outcomes, $\tilde{\mathbf{y}}_w$, corresponding to treatment $w$ and outputs a probability for each dosage level, $d_1^w, ..., d_{n_w}^w$, in a given realisation of $\tilde{\mathcal{D}}_w$. Writing $\mathbf{D}_w^j$ to denote the output of $\mathbf{D}_w$ corresponding to dosage level $D_j^w$, we define the loss of each dosage discriminator to be

$$\mathcal{L}_d(\mathbf{D}_w; \mathbf{G}) = -\mathbb{E}\left[\mathbb{I}_{\{W_f = w\}} \sum_{j=1}^{n_w} \mathbb{I}_{\{D_f = D_j^w\}} \log \mathbf{D}_w^j(\mathbf{X}, \tilde{\mathbf{Y}}_w) + \mathbb{I}_{\{D_f \neq D_j^w\}} \log(1 - \mathbf{D}_w^j(\mathbf{X}, \tilde{\mathbf{Y}}_w))\right], \tag{6}$$

where the expectation is taken over $\mathbf{X}, \tilde{\mathcal{D}}_w, \tilde{\mathbf{Y}}_w, W_f$ and $D_f$. The $\mathbb{I}_{\{W_f = w\}}$ term ensures that only samples for which the factual treatment is $w$ are used to train dosage discriminator $\mathbf{D}_w$ (otherwise there would be no factual dosage for that sample).

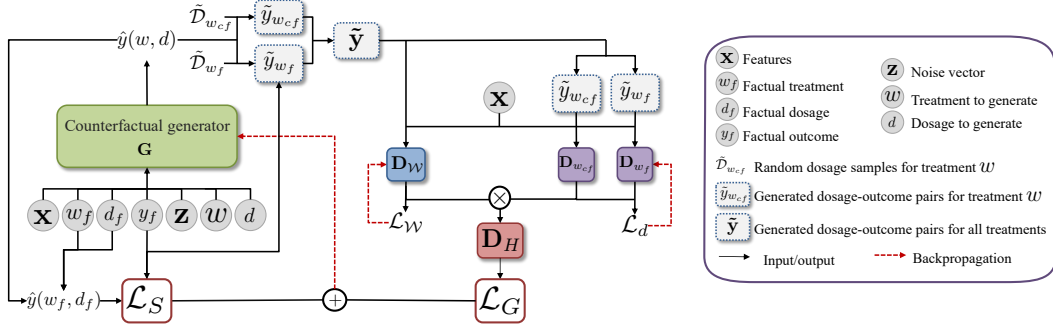

Figure 2: Overview of our model for the setting with two treatments ($w_f/w_{cf}$ being the factual/counterfactual treatment). The generator is used to generate an output for each dosage level in each $\tilde{\mathcal{D}}_w$, these outcomes together with the factual outcome, $y_f$, are used to create the set of dosage-outcome pairs, $\tilde{\mathbf{y}}$, which is passed to the treatment discriminator. Each dosage discriminator receives only the part of $\tilde{\mathbf{y}}$ corresponding to that treatment, i.e. $\tilde{\mathbf{y}}_w$. These discriminators are combined (Eq. 7) to define $\mathbf{D}_H$ which gives feedback to the generator.

We define the overall discriminator $\mathbf{D}_H : \mathcal{X} \times \prod_{w \in \mathcal{W}} (\mathcal{D}_w \times Y)^{n_w} \to [0,1]^{\sum n_w}$ by defining its output corresponding to the treatment-dosage pair $(w, d_j^w)$ as

$$\mathbf{D}_H^{w,j}(\mathbf{x}, \tilde{\mathbf{y}}) = \mathbf{D}_{\mathcal{W}}^w(\mathbf{x}, \tilde{\mathbf{y}}) \times \mathbf{D}_w^j(\mathbf{x}, \tilde{\mathbf{y}}_w). \tag{7}$$

Instead of the standard GAN minimax game, the generator and discriminators are trained according to the minimax game defined by seeking $\mathbf{G}^*, \mathbf{D}_H^*$ that solve:

$$\mathbf{G}^* = \arg\min_{\mathbf{G}} \mathcal{L}(\mathbf{D}_H^*; \mathbf{G}) + \lambda \mathcal{L}_S(\mathbf{G}) \qquad \mathbf{D}_H^{*\,w,j} = \mathbf{D}_{\mathcal{W}}^{*\,w} \times \mathbf{D}_w^{*\,j} \tag{8}$$

$$\mathbf{D}_{\mathcal{W}}^* = \arg\min_{\mathbf{D}_{\mathcal{W}}} \mathcal{L}_{\mathcal{W}}(\mathbf{D}_{\mathcal{W}}; \mathbf{G}^*) \qquad \mathbf{D}_w^* = \arg\min_{\mathbf{D}_w} \mathcal{L}_d(\mathbf{D}_w; \mathbf{G}^*), \forall w \in \mathcal{W} \tag{9}$$

Fig. 2 depicts our generator and hierarchical discriminator. Pseudo-code for our algorithm can be found in Appendix D.

### 4.3 Inference Network

Once we have learned the counterfactual generator, we can use it only to access (generated) dose-response curves for all samples in the dataset. To generate dose-response curves for a new sample we use the counterfactual generator along with the original data to train an inference network, $\mathbf{I} : \mathcal{X} \times \mathcal{T} \to \mathcal{Y}$. Details of the loss and pseudo-code can be found in Appendix E.

### 4.4 Theoretical Analysis

We now state our key theorem: the game defined by Eqs. (8-9) results in our hierarchical GAN learning counterfactuals that agree (in marginal distribution) with the true data.

**Theorem 1.** *The global minimum of* $\mathcal{L}(\mathbf{D}_H^*; \mathbf{G}) + \lambda \mathcal{L}_S(\mathbf{G})$ *subject to* $\mathbf{D}_H^{*\,w,j} = \mathbf{D}_{\mathcal{W}}^{*\,w} \times \mathbf{D}_w^{*\,j}$, $\mathbf{D}_{\mathcal{W}}^* = \arg\min_{\mathbf{D}_{\mathcal{W}}} \mathcal{L}_{\mathcal{W}}(\mathbf{D}_{\mathcal{W}}; \mathbf{G}^*)$ *and* $\mathbf{D}_w^* = \arg\min_{\mathbf{D}_w} \mathcal{L}_d(\mathbf{D}_w; \mathbf{G}^*), \forall w \in \mathcal{W}$ *is achieved if and only if for all* $\tilde{\mathcal{D}}_w$, *for all* $w, w' \in \mathcal{W}$ *and for all* $d \in \tilde{\mathcal{D}}_w, d' \in \tilde{\mathcal{D}}_{w'}$

$$p_{w,d}(\mathbf{y}|\mathbf{x}) = p_{w',d'}(\mathbf{y}|\mathbf{x}) \tag{10}$$

*which in turn implies that for any* $(w, d) \in \mathcal{T}$ *we have that the generated counterfactual for outcome* $(w, d)$ *for any sample (that was not assigned* $(w, d)$*) has the same (marginal) distribution (conditional on the features) as the true marginal distribution for that outcome.*

*Proof.* The proof and intermediate results can be found in Appendix C. $\qquad\square$

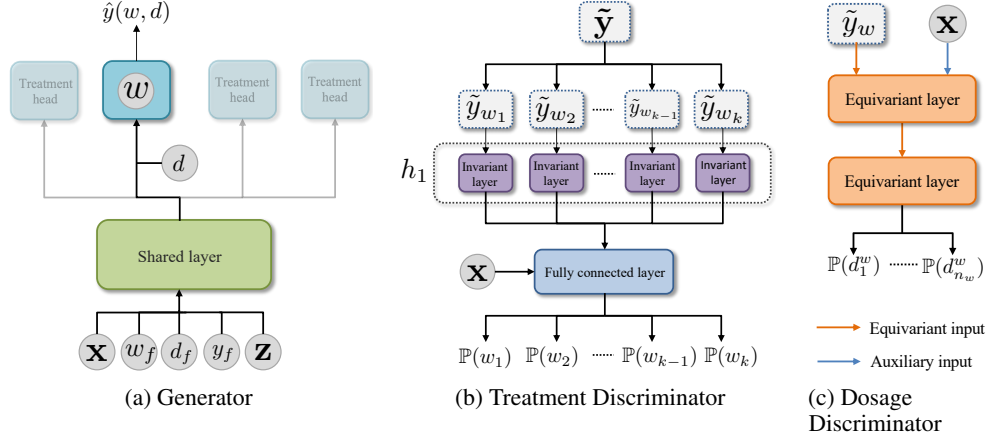

Figure 3: Architecture of our generator and discriminators.

# 5 Architecture

In this section, we describe in detail the novel architectures that we adopt to model each of the functions $\mathbf{G}, \mathbf{D}, \mathbf{D}_{\mathcal{W}}, \mathbf{D}_{w_1}, ..., \mathbf{D}_{w_k}$ which draws from the ideas in [15]. The inference network, $\mathbf{I}$, has the same architecture as the generator, but does not receive $w_f, d_f, y_f$ or $\mathbf{z}$ as inputs.

## 5.1 Generator Architecture

We adopt a multi-task deep learning model for $\mathbf{G}$ by defining a function $g : \mathcal{X} \times \mathcal{T} \times \mathcal{Y} \times \mathcal{Z} \to \mathcal{H}$ for some latent space $\mathcal{H}$ (typically $\mathbb{R}^l$ for some $l$) and then for each treatment $w \in \mathcal{W}$ we introduce a multitask "head", $g_w : \mathcal{H} \times \mathcal{D}_w \to \mathcal{Y}$ taking inputs from $\mathcal{H}$ *and* a dosage, $d$, to produce an outcome $\hat{y}(w, d) \in \mathcal{Y}$. Given observations, $(\mathbf{x}, t_f, y_f)$, a noise vector $\mathbf{z}$, and a target treatment-dosage pair, $t = (w, d)$, we define

$$\mathbf{G}(\mathbf{x}, t_f, y_f, \mathbf{z})(t) = g_w(g(\mathbf{x}, t_f, y_f, \mathbf{z}), d). \tag{11}$$

Each of $g, g_{w_1}, ..., g_{w_k}$ are fully connected networks. A figure of our generator architecture is given in Figure 3(a).

## 5.2 Hierarchical Discriminator Architectures

In order to ensure the discriminators act as functions of sets, we use ideas from [15] to create permutation invariant and permutation equivariant[5] networks. Zaheer et al. [15] provide several possible building blocks to use to construct invariant and equivariant deep networks. The basic building block we will use for invariant functions will be a layer of the form:

$$f_{inv}(\mathbf{u}) = \sigma(\mathbf{1}_b \mathbf{1}_m^T(\phi(u_1), ..., \phi(u_m))), \tag{12}$$

where $\mathbf{1}_l$ is a vector of 1s of dimension $l$, $\phi$ is any function $\phi : \mathcal{U} \to \mathbb{R}^q$ for some $q$ (in this paper we use a standard fully connected layer) and $\sigma$ is some non-linearity. The basic building block for equivariant functions is defined in terms of equivariance input, $\mathbf{u}$, and auxiliary input, $\mathbf{v}$, by:

$$f_{equi}(\mathbf{u}, \mathbf{v}) = \sigma(\lambda \mathbf{I}_m \mathbf{u} + \gamma(\mathbf{1}_m \mathbf{1}_m^T)\mathbf{u} + (\mathbf{1}_m \Theta^T)\mathbf{v}), \tag{13}$$

where $\mathbf{I}_m$ is the $m \times m$ identity matrix, $\lambda$ and $\gamma$ are scalar parameters and $\Theta$ is a vector of weights.

In the case of the hierarchical discriminator, we want the treatment discriminator, $\mathbf{D}_{\mathcal{W}}$, to be permutation invariant with respect to $\tilde{\mathbf{y}}_w$ for each treatment. To achieve this we define $h_1 : \prod_{w \in \mathcal{W}} (\mathcal{D}_w \times \mathcal{Y})^{n_w} \to \mathcal{H}_H$ and require that $h_1$ be permutation invariant w.r.t. each of the spaces $(\mathcal{D}_w \times \mathcal{Y})^{n_w}$. We concatenate the output of $h_1$ with the features $\mathbf{x}$ and pass these through a fully connected network $h_2 : \mathcal{X} \times \mathcal{H}_H \to [0, 1]^k$ so that $\mathbf{D}_{\mathcal{W}}(\mathbf{x}, \tilde{\mathbf{y}}) = h_2(\mathbf{x}, h_1(\tilde{\mathbf{y}}))$.

To construct $h_1$, we concatenate the outputs of several invariant layers of the form given in Eq. (12) that each individually act on the spaces $(\mathcal{D}_w \times \mathcal{Y})^{n_w}$. That is, for each treatment, $w \in \mathcal{W}$ we define a map $h_{inv}^w : (\mathcal{D}_w \times \mathcal{Y})^{n_w} \to \mathcal{H}_H^w$ by substituting $\tilde{\mathbf{y}}_w$ for $\mathbf{u}$ in Eq. (12). We then define $\mathcal{H}_H = \prod_{w \in \mathcal{W}} \mathcal{H}_H^w$ and $h_1(\tilde{\mathbf{y}}) = (h_{inv}^{w_1}(\tilde{\mathbf{y}}_{w_1}), ..., h_{inv}^{w_k}(\tilde{\mathbf{y}}_{w_k}))$.

We want each dosage discriminator, $\mathbf{D}_w$, to be permutation equivariant with respect to $\tilde{\mathbf{y}}_w$. To achieve this each $\mathbf{D}_w$ will consist of two layers of the form given in Eq. (13) with the equivariance input, $\mathbf{u}$, to the first layer being $\tilde{\mathbf{y}}_w$ and to the second layer being the output of the first layer and the auxiliary input, $\mathbf{v}$, to the first layer being the features, $\mathbf{x}$, and then no auxiliary input to the second layer.

Diagrams depicting the architectures of the treatment discriminator and dosage discriminators can be found in Fig. 3(b) and Fig. 3(c) respectively.

# 6 Evaluation

The nature of the treatment-effects estimation problem in even the binary treatments setting does not allow for meaningful evaluation on real-world datasets due to the inability to observe the counterfactuals. While there are well-established benchmark synthetic models for use in the binary (or multiple) case, no such models exist for the dosage setting. We propose our own semi-synthetic data simulation to evaluate our model against several benchmarks.

## 6.1 Experimental setup

**Semi-synthetic data generation:** We simulate data as follows. We obtain features, $\mathbf{x}$, from a real dataset (in this paper we use TCGA [22], News [19,23]) and MIMIC III [24])[6]. We consider 3 treatments each accompanied by a dosage. Each treatment, $w$, is associated with a set of parameters, $\mathbf{v}_1^w, \mathbf{v}_2^w, \mathbf{v}_3^w$. For each run of the experiment, these parameters are sampled randomly by sampling a vector, $\mathbf{u}_i^w$, from $\mathcal{N}(\mathbf{0}, \mathbf{1})$ and then setting $\mathbf{v}_i^w = \mathbf{u}_i^w / ||\mathbf{u}_i^w||$ where $|| \cdot ||$ is Euclidean norm. The shape of the response curve for each treatment, $f_w(\mathbf{x}, d)$, is given in Table 1, along with a closed-form expression for the optimal dosage. We add $\epsilon \sim \mathcal{N}(0, 0.2)$ noise to the outcomes.

We assign interventions by sampling a dosage, $d_w$, for each treatment from a beta distribution, $d_w|\mathbf{x} \sim \text{Beta}(\alpha, \beta_w)$. $\alpha \geq 1$ controls the dosage selection bias ($\alpha = 1$ gives the uniform distribution - see Appendix J). $\beta_w = \frac{\alpha-1}{d_w^*} + 2 - \alpha$, where $d_w^*$ is the optimal dosage[7] for treatment $w$. This setting of $\beta_w$ ensures that the mode of $\text{Beta}(\alpha, \beta_w)$ is $d_w^*$. We then assign a treatment according to $w_f|\mathbf{x} \sim \text{Categorical}(\text{softmax}(\kappa f(\mathbf{x}, d_w))$ where increasing $\kappa$ increases selection bias, and $\kappa = 0$ leads to random assignments. The factual intervention is given by $(w_f, d_{w_f})$. Unless otherwise specified, we set $\kappa = 2$ and $\alpha = 2$.

| Treatment | Dose-Response | Optimal dosage |
|---|---|---|
| 1 | $f_1(\mathbf{x}, d) = C((\mathbf{v}_1^1)^T \mathbf{x} + 12(\mathbf{v}_2^1)^T \mathbf{x} d - 12(\mathbf{v}_3^1)^T \mathbf{x} d^2)$ | $d_1^* = \frac{(\mathbf{v}_2^1)^T \mathbf{x}}{2(\mathbf{v}_3^1)^T \mathbf{x}}$ |
| 2 | $f_2(\mathbf{x}, d) = C((\mathbf{v}_1^2)^T \mathbf{x} + \sin(\pi(\frac{\mathbf{v}_2^2{}^T \mathbf{x}}{\mathbf{v}_3^2{}^T \mathbf{x}} d)))$ | $d_2^* = \frac{(\mathbf{v}_3^2)^T \mathbf{x}}{2(\mathbf{v}_2^2)^T \mathbf{x}}$ |
| 3 | $f_3(\mathbf{x}, d) = C((\mathbf{v}_1^3)^T \mathbf{x} + 12d(d-b)^2$, where $b = 0.75\frac{(\mathbf{v}_2^3)^T \mathbf{x}}{(\mathbf{v}_3^3)^T \mathbf{x}}$ | $\frac{b}{3}$ if $b \geq 0.75$ <br> $1$ if $b < 0.75$ |

Table 1: Dose response curves used to generate semi-synthetic outcomes for patient features $\mathbf{x}$. In the experiments, we set $C = 10$. $\mathbf{v}_1^w, \mathbf{v}_2^w, \mathbf{v}_3^w$ are the parameters associated with each treatment $w$.

**Benchmarks:** We compare against Generalized Propensity Score (GPS) [16] and Dose Reponse Networks (DRNet) [19] (the standard model and with Wasserstein regularization (DRN-W)). As a baseline, we compare against a standard multilayer perceptron (MLP) that takes patient features, treatment and dosage as input and estimates the patient outcome and a multitask variant (MLP-M) that has a designated head for each treatment. See Appendix K for details of the benchmark models and their hyperparameter optimisation. For metrics, we use Mean Integrated Square Error (MISE), Dosage Policy Error (DPE) and Policy Error (PE) [19,25]. For further details, see Appendix L.

## 6.2 Source of gain

Before comparing against the benchmarks, we investigate how each component of our model affects performance. We start with a baseline model in which both the generator and discriminator consist of a single fully connected network. One at a time, we add in the following components (cumulatively until we reach our full model): (1) the supervised loss in Eq. 3 ($+ \mathcal{L}_S$), (2) multitask heads in the generator (+ Multitask), (3) hierarchical discriminator (+ Hierarchical) and (4) invariance/equivariance layers in the treatment and dosage discriminators

|  | $\sqrt{\text{MISE}}$ | $\sqrt{\text{DPE}}$ | $\sqrt{\text{PE}}$ |
|---|---|---|---|
| Baseline | $4.18 \pm 0.32$ | $2.06 \pm 0.16$ | $1.93 \pm 0.12$ |
| $+ \mathcal{L}_S$ | $3.37 \pm 0.11$ | $1.14 \pm 0.05$ | $0.84 \pm 0.05$ |
| + Multitask | $3.15 \pm 0.12$ | $0.85 \pm 0.05$ | $0.67 \pm 0.05$ |
| + Hierchical | $2.54 \pm 0.05$ | $0.36 \pm 0.05$ | $0.45 \pm 0.05$ |
| + Inv/Eqv | $1.89 \pm 0.05$ | $0.31 \pm 0.05$ | $0.25 \pm 0.05$ |

Table 2: Source of gain analysis for our model on the TCGA. Metrics are reported as Mean $\pm$ Std.

(+Inv/Eqv). We report the results in Table 2 for TCGA for all 3 error metrics (MISE, DPE and PE), computed over 30 runs (results on News can be found in Appendix M).

The addition of each component results in improved performance, with the final row (our full model) demonstrating the best performance across both datasets and for all metrics. In Appendix M.2 we further compare our hierarchical discriminator with a single network discriminator by investigating both models sensitivity to the hyperparameter $n_w$. Details of the single discriminator can be found in Appendix G. Architectures for other components of the ablation studies can be found in Appendix H.

## 6.3 Benchmarks comparison

We now compare SCIGAN[8] against the benchmarks on our 3 semi-synthetic datasets. For MIMIC, due to the low number of samples available, we use two treatments - 2 and 3. We report $\sqrt{\text{MISE}}$ and $\sqrt{\text{PE}}$ in Table 3, $\sqrt{\text{DPE}}$ is given in Appendix M.3. We see that SCIGAN demonstrates a statistically significant improvement over every benchmark across all 3 datasets. In Appendix M.4 we compare SCIGAN with DRNET and GPS for an increasing number of treatments.

| Method | TCGA | | News | | MIMIC | |
|---|---|---|---|---|---|---|
|  | $\sqrt{\text{MISE}}$ | $\sqrt{\text{PE}}$ | $\sqrt{\text{MISE}}$ | $\sqrt{\text{PE}}$ | $\sqrt{\text{MISE}}$ | $\sqrt{\text{PE}}$ |
| SCIGAN | $\mathbf{1.89 \pm 0.05}$ | $\mathbf{0.25 \pm 0.05}$ | $\mathbf{3.71 \pm 0.05}$ | $\mathbf{3.90 \pm 0.05}$ | $\mathbf{2.09 \pm 0.12}$ | $\mathbf{0.32 \pm 0.05}$ |
| DRNet | $3.64 \pm 0.12$ | $0.67 \pm 0.05$ | $4.98 \pm 0.12$ | $4.17 \pm 0.11$ | $4.45 \pm 0.12$ | $1.44 \pm 0.05$ |
| DRN-W | $3.71 \pm 0.12$ | $0.63 \pm 0.05$ | $5.07 \pm 0.12$ | $4.56 \pm 0.12$ | $4.47 \pm 0.12$ | $1.37 \pm 0.05$ |
| GPS | $4.83 \pm 0.01$ | $1.60 \pm 0.01$ | $6.97 \pm 0.01$ | $24.1 \pm 0.05$ | $7.39 \pm 0.00$ | $20.2 \pm 0.01$ |
| MLP-M | $3.96 \pm 0.12$ | $1.20 \pm 0.05$ | $5.17 \pm 0.12$ | $5.82 \pm 0.16$ | $4.97 \pm 0.16$ | $1.59 \pm 0.05$ |
| MLP | $4.31 \pm 0.05$ | $0.97 \pm 0.05$ | $5.48 \pm 0.16$ | $6.45 \pm 0.21$ | $5.34 \pm 0.16$ | $1.65 \pm 0.05$ |

Table 3: Performance of individualized treatment-dose response estimation on three datasets. Bold indicates the method with the best performance for each dataset. Metrics are reported as Mean $\pm$ Std.

## 6.4 Discrete dosages

In this experiment, we investigate the discrete dosage setting. Details of the experimental setup can be found in Appendix M.6. We report Mean Squared Error (MSE) of SCIGAN and GANITE in Fig. 4 where we vary the number of discrete dosages from 3 to 30. We see that GANITE is incapable of handling more than 7 discrete dosages, whereas the hierarchical discriminator together with the invariant and equivariant layers allow SCIGAN to maintain performance as the number of dosages increases. Importantly, this demonstrates SCIGAN's wide-ranging applicability in both discrete and continuous settings.

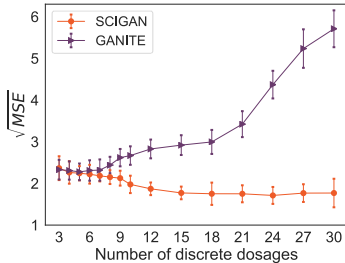

Figure 4: Comparison between SCIGAN and GANITE.

## 6.5 Treatment and dosage selection bias

Finally, we assess each model's robustness to treatment and dosage bias. We report $\sqrt{\text{MISE}}$ and $\sqrt{\text{PE}}$ on TCGA here. For the other metrics see Appendix M.8. Fig. 5(a) shows the performance of the 4 methods for $\kappa$ between $0$ (no bias) and $10$ (strong bias). Fig. 5(b) shows the performance for $\alpha$ between $1$ (no bias) and $8$ (strong bias). SCIGAN shows consistent performance, significantly outperforming the benchmarks for all $\kappa$ and $\alpha$.

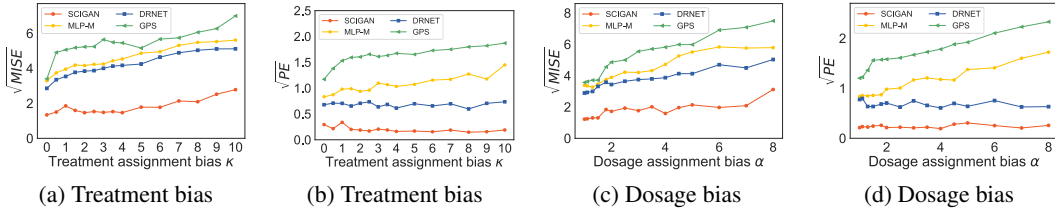

| (a) Treatment bias | (b) Treatment bias | (c) Dosage bias | (d) Dosage bias |

Figure 5: Performance of the 4 methods on datasets with varying bias levels.

## 7 Conclusion

In this paper we proposed a novel framework for estimating response curves for continuous interventions from observational data. We provided theoretical justification for our use of a modified GAN framework, which introduced a novel hierarchical discriminator. While our approach is very flexible (it accepts multiple treatments potentially with/without dosage parameters), one limitation is that SCIGAN needs at least a few thousand training samples, as is generally the case with neural networks, and GANs in particular. As for future research, we used out-of-the-box methods for the invariant and equivariant layers but feel that more can be done to make these layers as expressive as possible for this model. Moreover, another important research direction is to extend this work to the temporal causal inference setting [26–29] and estimate counterfactual outcomes for sequences of continuous interventions.

## Broader Impact

The impact of this problem in the healthcare setting is clear - being able to better estimate individualised responses to dosages will help us select treatments that result in improved patient outcomes. Moreover, clinicians and patients will often need to consider several different outcomes (such as potential side effects); better estimates of such outcomes allow the patients to make a more informed decision that is suitable for them.

Going beyond predictions by using causal inference methods to estimate the effect of interventions will result in more accurate and robust estimates and thus create more reliable components for use as part of decision support systems. Much of the recent work in causal inference has focused on binary or categorical treatments. Nevertheless, continuous interventions arise in many practical scenarios and building reliable methods for estimating their effects is paramount. We believe that our proposed model, SCIGAN, represents an important step forward in this direction. Nevertheless, we acknowledge the fact that the work presented in this paper is on the theoretical side and significant testing, potentially through clinical trials, will be needed before such methods can be used in practice, particularly due to the life-threatening implications of incorrect estimates. The risk of incorrectly assigning treatments can be significantly mitigated by ensuring that such models are used as *support* systems alongside clinicians, rather than instead of clinicians.

All work towards better estimating and understanding interventions can be used negatively, where someone wishing to cause harm can use the estimated outcomes to select the worst outcome.

## Acknowledgments

We would like to thank the reviewers for their valuable feedback. The research presented in this paper was supported by The Alan Turing Institute, under the EPSRC grant EP/N510129/1 and by the US Office of Naval Research (ONR), NSF 1722516.

## Footnotes

[3]In practice we set all $n_w$ to be the same. The default setting is 5 in the experiments.

[4]In practice, when $\mathcal{D}_w = [0, 1]$, each $D_j^w$ is sampled independently and uniformly from $[0, 1]$. Note that for each training iteration, $\tilde{\mathcal{D}}_w$ is resampled (see Section 1).

[5]Definitions can be found in [15] and in Appendix F.1.

[6]Details of each dataset can be found in Appendix I

[7]For symmetry, if $d_w^* = 0$, we sample $d_w$ from $1 - \text{Beta}(\alpha, \beta_w)$ where $\beta_w$ is set as though $d_w^* = 1$.

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
