[Supplementary Material]

# A Expanded Related Works

Most methods for performing causal inference in the static setting focus on the scenario with two or multiple treatment options and no dosage parameter. The approaches taken by such methods to estimate the treatment effects involve either building a separate regression model for each treatment [1, 30, 31] or using the treatment as a feature and adjusting for the imbalance between the different treatment populations. The former does not generalise to the dosage setting due to the now infinite number of possible treatments available. In the latter case, methods for handling the selection bias involve propensity weighting [2, 32, 33], building sub-populations using tree based methods [4, 5, 34, 35] or building balancing representations between patients receiving the different treatments [20, 23, 36, 37]. An additional approach involves modelling the data distribution of the factual and counterfactual outcomes [3, 6].

[25] leverages observational and interventional data to estimate the effects of discrete dosages for a single treatment. In particular, [25] uses observational data to construct a non-stationary covariance function and develop a hierarchical Gaussian process prior to build a distribution over the dose response curve. Then, controlled interventions are employed to learn a non-parametric affine transform to reshape this distribution. The setting in [25] differs significantly from ours as we do not assume access to any interventional data.

## A.1 Comparison with GANITE

Naive attempts to extend [6] to the continuous setting might involve: (1) discretising the continuous space of interventions; (2) somehow passing entire response curves to the discriminator and asking it to identify the point on the curve that corresponds to the factual outcome. Naturally, discretisation comes with a cost. If the discretisation is too coarse, the response curves will not be well-approximated. On the other hand, we show experimentally that GANITE is incapable of handling a high number of discrete interventions (corresponding to having a finer discretisation). In fact, although SCIGAN was designed for continuous interventions, it can be applied in the discrete setting and we show that it outperforms GANITE when the (discrete) parameter space is not small.

For (2), the problem is in defining a mechanism for generating these response curves in a form that can be passed to the discriminator and ensuring the *continuity* of these curves around the factual outcome so that the discontinuity itself does not make identification trivial for the discriminator. To overcome this we define a discriminator that acts on a finite set of points from each generated response curve (rather than on entire curves), as shown in Fig. 1. From *among the chosen points*, the discriminator attempts to identify the factual one. The set of points is sampled randomly *each* time an input would be passed to the discriminator. As our discriminator will be acting on a *set* of random intervention-outcome pairs, we explicitly condition it to behave as a function on a set. In particular, we draw on ideas from [15] to ensure that its output does not depend on the *order* of its input.

In addition, for the setting in which there are multiple possible interventions that *each* have an associated continuous parameter (which is the main setting of the paper), we propose a *hierarchical* discriminator which breaks down the job of the discriminator into determining the factual intervention and determining the factual parameter using separate networks. We show in the experiments section that this approach significantly improves performance and is more stable than using a single network discriminator. In this setting, we also model the generator as a multi-task deep network capable of taking a continuous parameter as an input; this gives us the flexibility to learn heterogeneous response curves for the different interventions.

## A.2 Off-policy evaluation and policy optimization with continuous treatments

A problem related to ours involves evaluating policies and learning optimal policies from logged data [38]. In this context, several methods have been proposed for performing off-policy evaluation and policy optimization with continuous treatments [39–41]. It should be emphasized that there is a significant difference between this line of research, which aims to find optimal policies, and the causal inference setting considered in this paper, where the aim is to learn the counterfactual patient outcomes under all possible treatment options.

For off-policy policy evaluation with continuous treatments, [39] propose a method based on inverse-propensity weighting to estimate the value function, i.e. cumulative reward of a target policy from

observational data. However, their proposed method does not perform any type of ITE estimation and does not have any intermediate per-sample value estimates. Alternatively, [41] does perform intermediate estimation of the value function for each sample. However, their contribution is to construct a doubly-robust estimator for the value function which crucially depends on knowing (or having some prior knowledge of) the parametric form of the response curve's dependency on the treatment. Without an assumed form, it is not possible to construct their estimator. Our proposed model, SCIGAN, does not assume any prior knowledge about the response curve.

The problem of individualised treatment effect estimation (which is our primary goal) is harder than off-policy evaluation as it involves estimating the outcomes (which may differ from the value function) for every sample and every possible action that could have been taken for that sample. After having learned these counterfactuals it is possible to perform policy evaluation, but note that the use cases for treatment effect estimation go beyond policy evaluation. In particular, for a given setting it may be far more beneficial to present a patient with their estimated outcome along with potential side effects and allow them (or the clinician) to come to a decision based on these several factors. In such a setting, each patient may have their own "internal" value function that depends on potential outcomes and side effects differently.

# B  Notation

In the table below, we summarise the notation used in our paper. Note that realisations of random variables are denoted using lowercase and subscripts/superscripts used with vector-valued functions denotes their output at the position of the given subscript/superscript.

| | |
|---|---|
| $\mathcal{X}$ | Feature space |
| $\mathcal{Y}$ | Outcome space |
| $\mathcal{T}$ | Intervention space |
| $\mathcal{W} = \{w_1, ..., w_k\}$ | Set of treatments |
| $\mathcal{D}_w$ | Dosage space for treatment $w \in \mathcal{W}$ |
| $\mathbf{X} \in \mathcal{X}$ | Features (random variable) |
| $Y : \mathcal{T} \to \mathcal{Y}$ | Potential outcome function (function-valued random variable) |
| $T_f = (W_f, D_f) \in \mathcal{T}$ | Factual/observed intervention (treatment-dosage pair) (random variable) |
| $Y_f \in \mathcal{Y}$ | Outcome corresponding to the observed intervention ($Y_f = Y(W_f, D_f)$) |
| $\mathbf{G}$ | Generator |
| $\mathbf{Z}$ | Random noise (for input to generator) (random variable) |
| $\hat{Y}_{cf} : \mathcal{T} \to \mathcal{Y}$ | Counterfactual outcome function induced by $\mathbf{G}$ |
| $\mathbf{D}$ | Discriminator |
| $\tilde{\mathcal{D}}_w = \{D_1^w, ..., D_{n_w}^w\}$ | Random (finite) subset of $\mathcal{D}_w$ |
| $n_w$ | Size of $\tilde{\mathcal{D}}_w$ (i.e. number of dosage levels passed to discriminator for treatment $w \in \mathcal{W}$) |
| $\tilde{\mathbf{Y}}_w = (D_i^w, \tilde{Y}_i^w)_{i=1}^{n_w}$ | Vector of dosage-outcome pairs generated by $\mathbf{G}$ (and $Y_f$) using $\tilde{\mathcal{D}}_w$ |
| $\mathbf{D}_{\mathcal{W}}$ | Treatment discriminator |
| $\mathbf{D}_w$ | Dosage discriminator for treatment $w \in \mathcal{W}$ |
| $\mathbf{D}_H$ | Hierarchical discriminator defined by combining $\mathbf{D}_{\mathcal{W}}$ and $\mathbf{D}_w$ |
| $\mathbf{I}$ | Inference network |
| $\mathcal{L}$ | GAN loss |
| $\mathcal{L}_S$ | Supervised loss |

# C   Proofs of Theoretical Results

In this section we prove Theorem 1. Note that these results also apply to GANITE (with very minor modifications - the proofs are even simpler in the case of GANITE).

In order to prove Theorem 1, we analyse the simpler minimax game defined by

$$\min_{\mathbf{G}} \max_{\mathbf{D}} \mathcal{L}(\mathbf{D}, \mathbf{G}) + \lambda \mathcal{L}_S(\mathbf{G}), \tag{14}$$

which corresponds to the single discriminator model (instead of the hierarchical model) and then use this to prove our full result.

**Lemma 1.** *Fix* $\mathbf{G}$ *and* $\tilde{\mathcal{D}} = \bigcup_w \tilde{\mathcal{D}}_w$. *Let* $p_{w,d}(\mathbf{y}|\mathbf{x}) = p_r(y_{w,d}|\mathbf{x})p_{\mathbf{G}}(\mathbf{y}_{\neg w,d}|\mathbf{x}, y_{w,d})$ *denote the induced joint density of outcomes when restricted to dosages in* $\tilde{\mathcal{D}}$, *where* $p_r$ *denotes the true density that generated the observed outcome and* $p_{\mathbf{G}}$ *denotes the density induced by* $\mathbf{G}$ *over the remaining dosages in* $\tilde{\mathcal{D}}$. *Then the optimal discriminator is*

$$\mathbf{D}^*_{w,j}(\mathbf{x}, \mathbf{y}) = \frac{\tilde{p}(w, d_j|\mathbf{x})p_{w,d_j}(\mathbf{y}|\mathbf{x})}{\sum_{w'\in\mathcal{W}} \sum_{i=1}^{n_w} \tilde{p}(w', d_i|\mathbf{x})p_{w',d_i}(\mathbf{y}|\mathbf{x})} \tag{15}$$

*where* $\tilde{p}$ *is the* $\tilde{\mathcal{D}}$-*restricted propensity given by* $\tilde{p}(w, d_j|\mathbf{x}) = p(w|\mathbf{x})(p(d_j|\mathbf{x}, w)/\sum_{i=1}^{n_w} p(d_i|\mathbf{x}, w))$.

*Proof.* Fix $\mathbf{G}$ and $\tilde{\mathcal{D}} = \bigcup_w \tilde{\mathcal{D}}_w$. The optimal discriminator is given by $\arg\min_{\mathbf{D}} \mathcal{L}(\mathbf{D}, \mathbf{G})$. We have

$$\mathcal{L}(\mathbf{D}, \mathbf{G}) = \mathbb{E}\left[ \sum_{w\in\mathcal{W}} \sum_{d\in\tilde{\mathcal{D}}_w} \mathbb{I}_{\{T_f=(w,d)\}} \log \mathbf{D}^{w,d}(\mathbf{X}, \tilde{\mathbf{Y}}) + \mathbb{I}_{\{T_f\neq(w,d)\}} \log(1 - \mathbf{D}^{w,d}(\mathbf{X}, \tilde{\mathbf{Y}})) \right] \tag{16}$$

$$= \mathbb{E}_{\tilde{\mathcal{D}}}\left[ \sum_{w\in\mathcal{W}} \sum_{d\in\tilde{\mathcal{D}}_w} \int_{(\mathbf{x},\mathbf{y})} \tilde{p}(w, d|\mathbf{x})p_{w,d}(\mathbf{y}|\mathbf{x}) \log \mathbf{D}^{w,d}(\mathbf{x}, \mathbf{y}) \right.$$
$$\left. + \left( \sum_{w',d'\neq w,d} \tilde{p}(w', d'|\mathbf{x})p_{w',d'}(\mathbf{y}|\mathbf{x}) \right) \log(1 - \mathbf{D}^{w,d}(\mathbf{x}, \mathbf{y}))p(\mathbf{x})d\mathbf{y}d\mathbf{x} \right] \tag{17}$$

where we have taken the (conditional on $\tilde{\mathcal{D}}$) expectations inside the sums and replaced indicator functions with densities as appropriate. We now note that $a \log p + b \log(1-p)$ for $p \in (0, 1)$ has a unique maximum at $p = \frac{a}{a+b}$, thus implying that the integrand is maximised when

$$\mathbf{D}^{w,d}(\mathbf{x}, \mathbf{y}) = \frac{\tilde{p}(w, d|\mathbf{x})p_{w,d}(\mathbf{y}|\mathbf{x})}{\tilde{p}(w, d|\mathbf{x})p_{w,d}(\mathbf{y}|\mathbf{x}) + \sum_{w',d'\neq w,d} \tilde{p}(w', d'|\mathbf{x})p_{w',d'}(\mathbf{y}|\mathbf{x})}. \tag{18}$$

This gives the required result.   $\square$

Using Lemma 1 we can now show that the optimal solution to our single discriminator game is when the marginal distributions of the generated counterfactuals are equal to the true counterfactuals. Importantly, this suffices for estimating $\mu$ since the expectation is only concerned with the marginal distribution of $Y(w, d)$.

**Lemma 2.** *The global minimum of the minimax game defined by* $\min_{\mathbf{G}} \max_{\mathbf{D}} \mathcal{L}(\mathbf{D}, \mathbf{G})$ *is achieved if and only if for all* $\tilde{\mathcal{D}}_w$, *for all* $w, w' \in \mathcal{W}$ *and for all* $d \in \tilde{\mathcal{D}}$, $d' \in \tilde{\mathcal{D}}_{w'}$ *we have that*

$$p_{w,d}(\mathbf{y}|\mathbf{x}) = p_{w',d'}(\mathbf{y}|\mathbf{x}) \tag{19}$$

*which in turn implies that for any treatment-dosage pair* $(w, d) \in \mathcal{T}$ *we have that the generated counterfactual for outcome* $(w, d)$ *for any sample (that was not assigned* $(w, d)$*) has the same (marginal) distribution (conditional on the features) as the true marginal distribution for that outcome.*

*Proof.* For fixed $\tilde{\mathcal{D}}$ and $\mathbf{x}$ we note that by substituting the optimal discriminator into $\mathcal{L}(\mathbf{D}, \mathbf{G})$ and subtracting $\sum_{w \in \mathcal{W}} \sum_{i=1}^{n_w} \log \tilde{p}(w, d_i | \mathbf{x})$ (which is independent of $\mathbf{G}$) we obtain

$$\mathcal{L}(\mathbf{D}^*, \mathbf{G}) - \int_{\mathbf{x}} \left( \sum_{w \in \mathcal{W}} \sum_{i=1}^{n_w} \log \tilde{p}(w, d_i | \mathbf{x}) \right) p(\mathbf{x}) d\mathbf{x} \tag{20}$$

$$= \mathbb{E}_{\tilde{\mathcal{D}}} \int_{\mathbf{x}} \mathrm{KL}\Big( p_{w,d}(\mathbf{y}|\mathbf{x}) || \hat{p}(\mathbf{y}|\mathbf{x}) \Big) + \mathrm{KL}\Big( \frac{1}{1 - \tilde{p}(w, d|\mathbf{x})} \sum_{t' \neq (w,d)} \tilde{p}(t'|\mathbf{x}) p_{t'}(\mathbf{y}|\mathbf{x}) || \hat{p}(\mathbf{y}|\mathbf{x}) \Big) d\mathbf{x} \tag{21}$$

where KL is the KL divergence and $\hat{p}(\mathbf{y}|\mathbf{x}) = \sum_{t \in \tilde{\mathcal{T}}} \tilde{p}(t|\mathbf{x}) p_t(\mathbf{y}|\mathbf{x})$ where $\tilde{\mathcal{T}}$ is the restriction of $\mathcal{T}$ to the dosages in $\tilde{\mathcal{D}}$. We then note that the KL divergence is minimised if and only if the two densities are equal, and we note by definition of $\hat{p}$ this occurs if and only if $p_{w,d}(\mathbf{y}|\mathbf{x}) = p_{w',d'}(\mathbf{y}|\mathbf{x})$ for all $w, d, w', d'$. This also directly implies that the marginal distributions for any fixed treatment-dosage pair agree for all factually observed treatments. In particular, if a sample received treatment $t' \neq t$, we have that the counterfactual generated for $t$ for this sample has the same distribution as the true data generating distribution. $\square$

Finally, we prove the following result, from which Theorem 1 follows immediately.

**Theorem 1.** *An optimal solution to the game defined by Equations 8 - 9 is also an optimal solution to the game defined by Equation 14 if the response curves generated by the generator for different treatments are conditionally independent given the features.*

*Proof.* To prove this result, it suffices to show that for fixed $\mathbf{G}$, $\mathbf{D}_H^* = \mathbf{D}^*$. To show this, we observe that by the same arguments as given for Lemma 1, we have the following:

$$\mathbf{D}_{\mathcal{W}}^{w\,*}(\mathbf{x}, \mathbf{y}) = \frac{p(w|\mathbf{x}) \Big( \sum_{i=1}^{n_w} \tilde{p}(d_i|\mathbf{x}, w) p_{w,d_i}(\mathbf{y}|\mathbf{x}) \Big)}{\sum_{w' \in \mathcal{W}} \Big( p(w'|\mathbf{x}) \sum_{i=1}^{n_w} \tilde{p}(d_i|\mathbf{x}, w) \Big)} \tag{22}$$

$$\mathbf{D}_w^{j\,*}(\mathbf{x}, \mathbf{y}_w) = \frac{\tilde{p}(d_j|\mathbf{x}, w) p_{w,d_j}(\mathbf{y}_w|\mathbf{x})}{\sum_{i=1}^{n_w} \tilde{p}(d_i|\mathbf{x}, w) p_{w,d_i}(\mathbf{y}_w|\mathbf{x})} \tag{23}$$

where $\mathbf{y}_w$ is the restriction of $\mathbf{y}$ to the outcomes corresponding to treatment $w$. By multiplying (23) by $\frac{p_{w,d_j}(\mathbf{y}_{\neq w}|\mathbf{y}, \mathbf{x})}{p_{w,d_j}(\mathbf{y}_{\neq w}|\mathbf{y}, \mathbf{x})}$ we obtain

$$\mathbf{D}_w^{j\,*}(\mathbf{x}, \mathbf{y}_w) = \frac{\tilde{p}(d_j|\mathbf{x}, w) p_{w,d_j}(\mathbf{y}|\mathbf{x})}{\sum_{i=1}^{n_w} \tilde{p}(d_i|\mathbf{x}, w) p_{w,d_i}(\mathbf{y}|\mathbf{x})} \tag{24}$$

since the conditional independence assumption implies that $p_{w,d_j}(\mathbf{y}_{\neq w}|\mathbf{y}, \mathbf{x}) = p_{w,d_i}(\mathbf{y}_{\neq w}|\mathbf{y}, \mathbf{x})$ for all $i, j = 1, ..., n_w$. Multiplying (22) and (24) together to get $\mathbf{D}_H^{w,j}$, we notice that the denominator in (24) cancels with the bracketed term of the numerator in (22) to give

$$\mathbf{D}_H^{*\,w,j} = \frac{p(w|\mathbf{x}) \tilde{p}(d_j|\mathbf{x}, w) p_{w,d_j}(\mathbf{y}|\mathbf{x})}{\sum_{w' \in \mathcal{W}} \Big( p(w'|\mathbf{x}) \sum_{i=1}^{n_w} \tilde{p}(d_i|\mathbf{x}, w) \Big)} \tag{25}$$

$$= \frac{\tilde{p}(w, d_j|\mathbf{x}) p_{w,d_j}(\mathbf{y}|\mathbf{x})}{\sum_{w' \in \mathcal{W}} \sum_{i=1}^{n_w} \tilde{p}(w', d_i|\mathbf{x}) p_{w',d_i}(\mathbf{y}|\mathbf{x})} \tag{26}$$

which is equal to the optimal discriminator for the single loss given in Lemma 1. $\square$

# D  Counterfactual Generator Pseudo-code

**Algorithm 1** Training of the generator in SCIGAN

---

1: **Input:** dataset $\mathcal{C} = \{(\mathbf{x}^i, t_f^i, y_f^i) : i = 1, ..., N\}$, batch size $n_{mb}$, number of dosages per treatment $n_d$, number of discriminator updates per iteration $n_D$, number of generator updates per iteration $n_G$, dimensionality of noise $n_z$, learning rate $\alpha$

2: **Initialize:** $\theta_G, \theta_{\mathcal{W}}, \{\theta_w\}_{w \in \mathcal{W}}$

3: **while G** has not converged **do**

   Discriminator updates

4:   **for** $i = 1, ..., n_D$ **do**

5:       Sample $(\mathbf{x}_1, (w_1, d_1), y_1), ..., (\mathbf{x}_{n_{mb}}, (w_{n_{mb}}, d_{n_{mb}}), y_{n_{mb}})$ from $\mathcal{C}$

6:       Sample generator noise $\mathbf{z}_j = (z_1^j, ..., z_{n_z}^j)$ from $\text{Unif}([0,1]^{n_z})$ for $j = 1, ..., n_{mb}$

7:       **for** $w \in \mathcal{W}$ **do**

8:           **for** $j = 1, ..., n_{mb}$ **do**

9:               Sample $\tilde{D}_w^j = (d_1^{w,j}, ..., d_{n_d}^{w,j})$ independently and uniformly from $(\mathcal{D}_w)^{n_d}$

10:              Set $\tilde{\mathbf{y}}_w^j$ according to Eq. 4

11:              Calculate gradient of dosage discriminator loss

$$g_w \leftarrow \nabla_{\theta_w} - \left[ \sum_{\{j:w_j=w\}} \sum_{k=1}^{n_d} \mathbb{I}_{\{d_j = d_k^{w,j}\}} \log \mathbf{D}_w(\mathbf{x}_j, \tilde{\mathbf{y}}_w^j) + \mathbb{I}_{\{d_j \neq d_k^{w,j}\}} \log(1 - \mathbf{D}_w(\mathbf{x}_j, \tilde{\mathbf{y}}_w^j)) \right]$$

12:              Update dosage discriminator parameters $\theta_w \leftarrow \theta_w + \alpha g_w$

13:          Set $\tilde{\mathbf{y}}_j = (\tilde{\mathbf{y}}_w^j)_{w \in \mathcal{W}}$

14:          Calculate gradient of treatment discriminator loss

$$g_{\mathcal{W}} \leftarrow \nabla_{\theta_{\mathcal{W}}} - \left[ \sum_{j=1}^{n_{mb}} \sum_{w \in \mathcal{W}} \mathbb{I}_{\{w_j = w\}} \log \mathbf{D}_{\mathcal{W}}(\mathbf{x}_j, \tilde{\mathbf{y}}_j) + \mathbb{I}_{\{w_j \neq w\}} \log(1 - \mathbf{D}_{\mathcal{W}}(\mathbf{x}_j, \tilde{\mathbf{y}}_j)) \right]$$

15:          Update treatment discriminator parameters $\theta_{\mathcal{W}} \leftarrow \theta_{\mathcal{W}} + \alpha g_{\mathcal{W}}$

16:   Generator updates

17:   **for** $i = 1, ..., n_G$ **do**

18:       Sample $(\mathbf{x}_1, (w_1, d_1), y_1), ..., (\mathbf{x}_{n_{mb}}, (w_{n_{mb}}, d_{n_{mb}}), y_{n_{mb}})$ from $\mathcal{C}$

19:       Sample generator noise $\mathbf{z}_j = (z_1^j, ..., z_{n_z}^j)$ from $\text{Unif}([0,1]^{n_z})$ for $j = 1, ..., n_{mb}$

20:       Sample $(\tilde{D}_w^j)_{w \in \mathcal{W}}$ from $\Pi_{w \in \mathcal{W}}(\mathcal{D}_w)^{n_d}$ for $j = 1, ..., n_{mb}$

21:       Set $\tilde{\mathbf{y}}$ according to Eq. 4

22:       Calculate gradient of generator loss

$$g_G \leftarrow \nabla_{\theta_G} \left[ \sum_{j=1}^{n_{mb}} \sum_{w \in \mathcal{W}} \sum_{l=1}^{n_d} \mathbb{I}_{\{w_j = w, d_j = d_l^{w,j}\}} \log(\mathbf{D}_{\mathcal{W}}^w(\mathbf{x}_j, \tilde{\mathbf{y}}_j)_w \times \mathbf{D}_w^l(\mathbf{x}_j, \tilde{\mathbf{y}}_w^j)_l) \right.$$

$$\left. + \mathbb{I}_{\{w_j \neq w, d_j \neq d_l^{w,j}\}} \log(1 - (\mathbf{D}_{\mathcal{W}}^w(\mathbf{x}_j, \tilde{\mathbf{y}}_j) \times D_w^l(\mathbf{x}_j, \tilde{\mathbf{y}}_w^j))) \right]$$

23:       Update generator parameters $\theta_G \leftarrow \theta_G + \alpha g_G$

24: **Output: G**

---

# E   Inference Network

To generate dose-response curves for new samples, we learn an inference network, $\mathbf{I} : \mathbf{X} \times \mathcal{T} \rightarrow \mathcal{Y}$. This inference network is trained using the original dataset and the learned counterfactual generator. As with the training of the generator and discriminator, we train using a random set of dosages, $\tilde{\mathcal{D}}_w$. The loss is given by

$$\mathcal{L}_I(\mathbf{I}) = \mathbb{E}\left[ \sum_{w\in\mathcal{W}} \sum_{d\in\tilde{\mathcal{D}}_w} (\tilde{Y}(w,d) - \mathbf{I}(\mathbf{X},(w,d)))^2 \right], \tag{27}$$

where $\tilde{Y}(w,d)$ is $Y_f$ if $T_f = (w,d)$ or given by the generator if $T_f \neq (w,d)$. The expectation is taken over $\mathbf{X}, T_f, Y_f, \mathbf{Z}$ and $\tilde{\mathcal{D}}_w$.

## E.1   Pseudo-code for training the Inference Network

---
**Algorithm 2** Training of the inference network in SCIGAN

---
1: **Input:** dataset $\mathcal{C} = \{(\mathbf{x}^i, t_f^i, y_f^i) : i = 1, ..., N\}$, trained generator $\mathbf{G}$, batch size $n_{mb}$, number of dosages per treatment $n_d$, dimensionality of noise $n_z$, learning rate $\alpha$
2: **Initialize:** $\theta_I$
3: **while I** has not converged **do**
4:     Sample $(\mathbf{x}_1, (w_1, d_1), y_1), ..., (\mathbf{x}_{n_{mb}}, (w_{n_{mb}}, d_{n_{mb}}), y_{n_{mb}})$ from $\mathcal{C}$
5:     Sample generator noise $\mathbf{z}_j = (z_1^j, ..., z_{n_z}^j)$ from $\text{Unif}([0,1]^{n_z})$ for $j = 1, ..., n_{mb}$
6:     **for** $j = 1, ..., n_{mb}$ **do**
7:         **for** $w \in \mathcal{W}$ **do**
8:             Sample $\tilde{D}_w^j = (d_1^{w,j}, ..., d_{n_d}^{w,j})$ independently and uniformly from $(\mathcal{D}_w)^{n_d}$
9:             Set $\tilde{\mathbf{y}}_w^j$ according to Eq. 4 4
10:     Calculate gradient of inference network loss

$$g_I \leftarrow -\nabla_{\theta_I}\left[ \sum_{j=1}^{n_{mb}} \sum_{w\in\mathcal{W}} \sum_{l=1}^{n_d} (\tilde{\mathbf{y}}_w^j)_l - \mathbf{I}(\mathbf{x}_j, (w, d_l^{w,j}))^2 \right]$$

11:     Update inference network parameters $\theta_I \leftarrow \theta_I + \alpha g_I$
12: **Output: I**

---

# F  Architecture

## F.1  Definitions of Permutation Invariance and Permutation Equivariance

The notions of what it means for a function to be *permutation invariant* and *permutation equivariant* with respect to (a subset of) its inputs are given below in definitions 1 and 2, respectively. Let $\mathcal{U}, \mathcal{V}, \mathcal{C}$ be some spaces. Let $m \in \mathbb{Z}^+$.

**Definition 1.** *A function $f : \mathcal{U}^m \times \mathcal{V} \to \mathcal{C}$ is permutation* invariant *with respect to the space $\mathcal{U}^m$ if for every $\mathbf{u} = (u_1, ..., u_m) \in \mathcal{U}^m$, every $v \in \mathcal{V}$ and every permutation, $\sigma$, of $\{1, ..., m\}$ we have*

$$f(u_1, ..., u_m, v) = f(u_{\sigma(1)}, ..., u_{\sigma(m)}, v) \,. \tag{28}$$

**Definition 2.** *A function $f : \mathcal{U}^m \times \mathcal{V} \to \mathcal{C}^m$ is permutation* equivariant *with respect to the space $\mathcal{U}^m$ if for every $\mathbf{u} \in \mathcal{U}^m$, every $v \in \mathcal{V}$ and every permutation, $\sigma$, of $\{1, ..., m\}$ we have $f(u_{\sigma(1)}, ..., u_{\sigma(m)}, v) = (f_{\sigma(1)}(\mathbf{u}, v), ..., f_{\sigma(m)}(\mathbf{u}, v))$, where $f_j(\mathbf{u}, v)$ is the jth element of $f(\mathbf{u}, v)$.*

To build up functions that are permutation invariant and permutation equivariant we make the following observations: (1) the composition of any function with a permutation invariant function is permutation invariant, (2) the composition of two permutation equivariant functions is permutation equivariant.

As noted in Section 5.2, the basic building block we use for equivariant functions is defined in terms of equivariance input, $\mathbf{u}$, and auxiliary input, $\mathbf{v}$ by

$$f_{equi}(\mathbf{u}, \mathbf{v}) = \sigma(\lambda \mathbf{I}_m \mathbf{u} + \gamma (\mathbf{1}_m \mathbf{1}_m^T) \mathbf{u} + (\mathbf{1}_m \Theta^T) \mathbf{v}) \,. \tag{29}$$

# G  Single Discriminator Model

In the paper we developed a hierarchical discriminator and demonstrated that it performs significantly better than the single discriminator setup that we now describe in this section.

## G.1  Single Discriminator

In the single model, we will aim to learn a single discriminator, $\mathbf{D}$, that outputs $\mathbb{P}((W_f, D_f) = (w,d)|\mathbf{X}, \tilde{\mathcal{D}}_w, \tilde{\mathbf{Y}})$ for each $w \in \mathcal{W}$ and $d \in \tilde{\mathcal{D}}_w$. We will write $\mathbf{D}^{w,d}(\cdot)$ to denote the output of $\mathbf{D}$ that corresponds to the treatment-dosage pair $(w,d)$. We define the loss, $\mathcal{L}_D$, to be

$$\mathcal{L}_D(\mathbf{D}; \mathbf{G}) = -\mathbb{E}\left[\sum_{w \in \mathcal{W}} \sum_{d \in \tilde{\mathcal{D}}_w} \mathbb{I}_{\{T_f = (w,d)\}} \log \mathbf{D}^{w,d}(\mathbf{X}, \tilde{\mathbf{Y}}) + \mathbb{I}_{\{T_f \neq (w,d)\}} \log(1 - \mathbf{D}^{w,d}(\mathbf{X}, \tilde{\mathbf{Y}}))\right] \tag{30}$$

where the expectation is taken over $\mathbf{X}, \{\tilde{\mathcal{D}}_w\}_{w \in \mathcal{W}}, \tilde{\mathbf{Y}}, W_f$ and $D_f$ and we note that the dependence on $\mathbf{G}$ is through $\tilde{\mathbf{Y}}$. Our single discriminator will be trained to minimise this loss directly. The generator GAN-loss, $\mathcal{L}_G$, is then defined by

$$\mathcal{L}_G(\mathbf{G}) = -\mathcal{L}_D(\mathbf{D}^*; \mathbf{G}) \tag{31}$$

where $\mathbf{D}^*$ is the optimal discriminator given by minimising $\mathcal{L}_D$. The generator will be trained to minimise $\mathcal{L}_G + \lambda \mathcal{L}_S$.

## G.2  Single Discriminator Architecture

In the case of the single discriminator, we want the output of $\mathbf{D}$ corresponding to each treatment $w \in \mathcal{W}$, i.e. $(\mathbf{D}^{w,1}, ..., \mathbf{D}^{w,n_w})$, to be permutation equivariant with respect to $\tilde{\mathbf{y}}_w$ and permutation invariant with respect to each $\tilde{\mathbf{y}}_v$ for $v \in \mathcal{W} \setminus \{w\}$. To achieve this, we first define a function $f : \prod_{w \in \mathcal{W}} (\mathcal{D}_w \times \mathcal{Y})^{n_w} \to \mathcal{H}_S$ and require that this function be permutation invariant with respect to each of the spaces $(\mathcal{D}_w \times \mathcal{Y})^{n_w}$. For each treatment, $w \in \mathcal{W}$, we introduce a multi-task head, $f_w : \mathcal{X} \times \mathcal{H}_S \times (\mathcal{D}_w \times \mathcal{Y})^{n_w} \to [0,1]^{n_w}$, and require that each of these functions be permutation equivariant with respect to their corresponding input space $(\mathcal{D}_w \times \mathcal{Y})^{n_w}$ but they can depend on the features, $\mathbf{x} \in \mathcal{X}$, and invariant latent representation coming from $f$ arbitrarily. Writing $f_w^j$ to denote the $j$th output of $f_w$, the output of the discriminator given input features, $\mathbf{x}$, and generated outcomes, $\tilde{\mathbf{y}}$, is defined by

$$\mathbf{D}^{w,j}(\mathbf{x}, \tilde{\mathbf{y}}) = f_w^i(\mathbf{x}, f(\tilde{\mathbf{y}}), \tilde{\mathbf{y}}_w). \tag{32}$$

Figure 6: Overview of the single discriminator architecture.

To construct the function $f$, we concatenate the outputs of several invariant layers of the form given in Eq. 29 that each individually act on the spaces $(\mathcal{D}_w \times \mathcal{Y})^{n_w}$. That is, for each treatment, $w \in \mathcal{W}$ we define a map $f_{inv}^w : (\mathcal{D}_w \times \mathcal{Y})^{n_w} \to \mathcal{H}_S^w$ by substituting $\tilde{\mathbf{y}}_w$ for $\mathbf{u}$ in Eq. 29. We then define $\mathcal{H}_S = \prod_{w \in \mathcal{W}} \mathcal{H}_S^w$ and $f(\tilde{\mathbf{y}}) = (f_{inv}^{w_1}(\tilde{\mathbf{y}}_{w_1}), ..., f_{inv}^{w_k}(\tilde{\mathbf{y}}_{w_k}))$.

Each $f_w$ will consist of two layers of the form given in Eq. (29) with the equivariance input, $\mathbf{u}$, to first layer being $\tilde{\mathbf{y}}_w$ and to the second layer being the output of the first layer and the auxiliary input, $\mathbf{v}$, to the first layer being the concatenation of the features and invariant representation, i.e. $(\mathbf{x}, f(\tilde{\mathbf{y}}))$ and then no auxiliary input to the second layer.

A diagram depicting the architecture of the single discriminator model can be found in Fig. 6.

# H Ablation Studies architectures

(a) Generator without multitask heads

(b) Treatment discriminator with fully connected (FC) layers instead of invariant layers

(c) Dosage discriminator with FC layers instead of equivariant layers

Figure 7: Architecture of the generator without multitask and discriminator without the invariant/equivariant layers used in the ablation studies.

# I Dataset descriptions

**TCGA:** The TCGA dataset consists of gene expression measurements for cancer patients [22]. There are 9659 samples for which we used the measurements from the 4000 most variable genes. The gene expression data was log-normalized and each feature was scaled in the $[0, 1]$ interval. For each patient, the features were scaled to have norm 1. We give meaning to our treatments and dosages by considering the treatment as being chemotherapy/radiotherapy/immunotherapy and their corresponding dosages. The outcome can be thought of as the risk of cancer recurrence [19]. We used the same version of the TCGA dataset as used by DRNet https://github.com/d909b/drnet.

**News:** The News dataset consists of word counts for news items. We extracted 10000 news items (randomly sampled) each with 2858 features. As in [19,23], we give meaning to our treatments and dosages by considering the treatment as being the viewing device (e.g. phone, tablet etc.) used to read the article and the dosage as being the amount of time spent reading it. The outcome can be thought of as user satisfaction. We used the same version of the News dataset as used by DRNet https://github.com/d909b/drnet.

**MIMIC III:** The Medical Information Mart for Intensive Care (MIMIC III) [24] database consists of observational data from patients in the ICU. We extracted 3000 patients (randomly sampled) that receive antibiotics treatment and we used as features 9 clinical covariates, namely age, temperature, heart rate, systolic and diastolic blood pressure, SpO2, FiO2, glucose, and white blood cell count, measured at start of ICU stay. Again, the features were scaled in the $[0, 1]$ interval. In this setting, we can considered as treatments the different antibiotics and their corresponding dosages.

For a summary description of the datasets, see table 4. The datasets are split into 64/16/20% for training, validation and testing respectively. The validation dataset is used for hyperparameter optimization.

|                     | TCGA | News  | MIMIC |
|---------------------|------|-------|-------|
| Number of samples   | 9659 | 10000 | 3000  |
| Number of features  | 4000 | 2858  | 9     |
| Number of treatments | 3*  | 3     | 2     |

Table 4: Summary description of datasets. *: for our final experiment in Appendix M.4 we increase the number of treatments in TCGA to 6 and 9.

## J  Dosage bias

In order to create dosage-assignment bias in our datasets, we assign dosages according to $d_w|\mathbf{x} \sim$ $\text{Beta}(\alpha, \beta_w)$. The selection bias is controlled by the parameter $\alpha \geq 1$. When we set $\beta_w = \frac{\alpha-1}{d_w^*} + 2 - \alpha$ (which ensures that the mode of our distribution is $d_w^*$), we can write the variance of $d_w$ in terms of $\alpha$ and $d_w^*$ as follows

$$\text{Var}(d_w) = \frac{\frac{\alpha^2 - \alpha}{d_w^*} + 2\alpha - \alpha^2}{(\frac{\alpha-1}{d_w^*} + 2)^2(\frac{\alpha-1}{d_w^*} + 3)} \approx \frac{c\alpha^2}{d\alpha^3} \ . \tag{33}$$

We see that the variance of our Beta distribution therefore decreases with $\alpha$, resulting in the sampled dosages being closer to the optimal dosage, thus resulting in higher dosage-selection bias. In addition we note that the $\text{Beta}(1, 1)$ distribution is in fact the uniform distribution, corresponding to the dosages being sampled independently of the patient features, resulting in no selection bias when $\alpha = 1$.

# K  Benchmarks

We use the publicly available GitHub implementation of DRNet provided by [19]: `https://github.com/d909b/drnet`. Moreover, we also used a GPS implementation similar to the one from `https://github.com/d909b/drnet` which uses the `causaldrf` R package [42]. More spcifically, the GPS implementation uses a normal treatment model, a linear treatment formula and a 2-nd degree polynomial for the outcome. Moreover, for the TCGA and News datasets, we performed PCA and only used the 50 principal components as input to the GPS model to reduce computational complexity.

**Hyperparameter optimization:** The validation split of the dataset is used for hyperparameter optimization. For the DRNet benchmarks we use the same hyperparameter optimization proposed by [19] with the hyperparameter search ranges described in Table 5. For SCIGAN, we use the hyperparameter optimization method proposed in GANITE [6], where we use the complete dataset from the counterfactual generator to evaluate the MISE on the inference network. We perform a random search [43] for hyperparameter optimization over the search ranges in Table 6. For all experiments with SCIGAN, we used 5000 training iterations for the GAN network and 10000 training iterations for the inference network. This number of training iterations was chosen to ensure convergence of the generator loss, discriminator loss, as well as of the supervised loss. For a fair comparison, for the MLP-M model we used the same architecture used in the inference network of SCIGAN. Similarly, for the MLP model we use the same architecture as for the MLP-M, but without the multitask heads.

| Hyperparameter | Search range |
|---|---|
| Batch size | 32, 64, 128 |
| Number of units per hidden layer | 24, 48, 96, 192 |
| Number of hidden layers | 2, 3 |
| Dropout percentage | 0.0, 0.2 |
| Imbalance penalty weight$^*$ | 0.1, 1.0, 10.0 |
| | Fixed |
| Number of dosage strata $E$ | 5 |

Table 5: Hyperparameters search range for DRNet.  *: For the DRNet model using Wasserstein regularization only.

| Hyperparameter | Search range |
|---|---|
| Batch size | 64, 128, 256 |
| Number of units per hidden layer | 32, 64, 128 |
| Size of invariant and equivariant representations | 16, 32, 64, 128 |
| | Fixed |
| Number of hidden layers per multitask head | 2 |
| Number of dosage samples | 5 |
| $\lambda$ | 1 |
| Optimization | Adam Moment Optimization |

Table 6: Hyperparameters search range for SCIGAN.

The hyperparameters used to generate the results for SCIGAN are given in Table 7.

The experiments were run on a system with 6CPUs, an Nvidia K80 Tesla GPU and 56GB of RAM.

| Hyperparameter | TCGA | News | MIMIC |
|---|---|---|---|
| Batch size | 128 | 256 | 128 |
| Number of units per hidden layer | 64 | 128 | 32 |
| Size of invariant and equivariant representations | 16 | 32 | 16 |
| Number of hidden layers per multitask head | 2 | 2 | 2 |
| Number of dosage samples | 5 | 5 | 5 |
| $\lambda$ | 1 | 1 | 1 |

Table 7: Hyperparameters used for obtaining results.

## L    Metrics

The Mean Integrated Square Error (MISE) measures how well the models estimates the patient outcome across the entire dosage space:

$$\text{MISE} = \frac{1}{N}\frac{1}{k}\sum_{w\in\mathcal{W}}\sum_{i=1}^{N}\int_{\mathcal{D}_w}\left(y^i(w,u) - \hat{y}^i(w,u)\right)^2 du\,. \tag{34}$$

In addition to this, we also compute the mean dosage policy error (DPE) [19] to assess the ability of the model to estimate the optimal dosage point for every treatment for each individual:

$$\text{DPE} = \frac{1}{N}\frac{1}{k}\sum_{w\in\mathcal{W}}\sum_{i=1}^{N}\left(y^i(w,d_w^*) - y^i(w,\hat{d}_w^*)\right)^2\,, \tag{35}$$

where $d_w^*$ is the true optimal dosage and $\hat{d}_w^*$ is the optimal dosage identified by the model. The optimal dosage points for a model are computed using SciPy's implementation of Sequential Least SQuares Programming.

Finally, we compute the mean policy error (PE) [19] which compares the outcome of the true optimal treatment-dosage pair to the outcome of the optimal treatment-dosage pair as selected by the model:

$$\text{PE} = \frac{1}{N}\sum_{i=1}^{N}\left(y^i(w^*,d_w^*) - y^i(\hat{w}^*,\hat{d}_w^*)\right)^2\,, \tag{36}$$

where $w^*$ is the true optimal treatment and $\hat{w}^*$ is the optimal treatment identified by the model. The optimal treatment-dosage pair for a model is selected by first computing the optimal dosage for each treatment and then selecting the treatment with the best outcome for its optimal dosage.

Each of these metrics are computed on a held out test-set.

# M Additional results

## M.1 News results for source of gain

|  | **News** | | |
|  | $\sqrt{\mathrm{MISE}}$ | $\sqrt{\mathrm{DPE}}$ | $\sqrt{\mathrm{PE}}$ |
| --- | --- | --- | --- |
| Baseline | $6.17 \pm 0.27$ | $6.97 \pm 0.27$ | $6.20 \pm 0.21$ |
| $+ \mathcal{L}_S$ | $4.51 \pm 0.16$ | $4.46 \pm 0.12$ | $4.40 \pm 0.11$ |
| + Multitask | $4.11 \pm 0.11$ | $4.33 \pm 0.11$ | $4.31 \pm 0.11$ |
| + Hierarchical | $4.07 \pm 0.05$ | $4.24 \pm 0.11$ | $4.17 \pm 0.12$ |
| + Inv/Eqv | $3.71 \pm 0.05$ | $4.14 \pm 0.11$ | $3.90 \pm 0.05$ |

Table 8: Source of gain analysis for our model on the News dataset. Metrics are reported as Mean $\pm$ Std.

## M.2 Investigating hyperparameter sensitivity ($n_w$)

The performance of the single discriminator causes significant performance drops around $n_w = 9$ across all metrics. As previously noted, this is due to the dimension of the output space (which for $n_w = 9$ is 27) being too large. Conversely, we see that our hierarchical discriminator shows much more stable performance even when $n_w = 19$.

Here we present additional results for our investigation of the hyperparameters $n_w$. Fig. 8 reports each of the 3 performance metrics as we increase the number of dosage samples, $n_w$, used to train the discriminators on the News dataset. As with the TCGA results in the main paper we see that the single discriminator suffers a significant performance decrease when $n_w$ is set too high.

(a) $\sqrt{\mathrm{MISE}}$     (b) $\sqrt{\mathrm{DPE}}$     (c) $\sqrt{\mathrm{PE}}$

Figure 8: Performance of single vs. hierarchical discriminator when increasing the number of dosage samples ($n_w$) on News dataset.

(a) $\sqrt{\mathrm{MISE}}$     (b) $\sqrt{\mathrm{DPE}}$     (c) $\sqrt{\mathrm{PE}}$

Figure 9: Performance of single vs. hierarchical discriminator when increasing the number of dosage samples ($n_w$) on TCGA dataset.

## M.3  Dosage Policy Error for Benchmark Comparison

In Table 9 we report the Dosage Policy Error (DPE) corresponding to Section 6.3 in the main paper.

| Methods | TCGA $\sqrt{\text{DPE}}$ | News $\sqrt{\text{DPE}}$ | MIMIC $\sqrt{\text{DPE}}$ |
|---|---|---|---|
| **SCIGAN** | $\mathbf{0.31} \pm 0.05$ | $\mathbf{4.14} \pm 0.11$ | $\mathbf{0.51} \pm 0.05$ |
| DRNet | $0.51 \pm 0.05^*$ | $4.39 \pm 0.11^*$ | $0.52 \pm 0.05$ |
| DRN-W | $0.50 \pm 0.05^*$ | $4.21 \pm 0.11$ | $0.53 \pm 0.05$ |
| GPS | $1.38 \pm 0.01^*$ | $6.40 \pm 0.01^*$ | $1.41 \pm 0.12^*$ |
| MLP-M | $0.92 \pm 0.05^*$ | $4.94 \pm 0.16^*$ | $0.77 \pm 0.05^*$ |
| MLP | $1.04 \pm 0.05^*$ | $5.18 \pm 0.12^*$ | $0.80 \pm 0.05^*$ |

Table 9: Performance of individualized treatment-dose response estimation on three datasets. Bold indicates the method with the best performance for each dataset. *: performance improvement is statistically significant.

## M.4  Varying the number of treatments

In this experiment, we increase the number of treatments by defining 3 or 6 additional treatments. The parameters $\mathbf{v}_1^w, \mathbf{v}_2^w, \mathbf{v}_3^w$ are defined in exactly the same way as for 3 treatments. The outcome shapes for treatments 4 and 7 are the same as for treatment 1, similarly for 5, 8 and 2 and for 6, 9 and 3. In Table 10 we report MISE, DPE and PE on the TCGA dataset with 6 treatments (TCGA-6) and with 9 treatments (TCGA-9). Note that we use 3 dosage samples for training SCIGAN in this experiment.

| Method | TCGA - 6 $\sqrt{\text{MISE}}$ | $\sqrt{\text{DPE}}$ | $\sqrt{\text{PE}}$ | TCGA - 9 $\sqrt{\text{MISE}}$ | $\sqrt{\text{DPE}}$ | $\sqrt{\text{PE}}$ |
|---|---|---|---|---|---|---|
| SCIGAN | $2.37 \pm 0.12$ | $0.43 \pm 0.05$ | $0.32 \pm 0.05$ | $2.79 \pm 0.05$ | $0.51 \pm 0.05$ | $0.54 \pm 0.05$ |
| DRNET | $4.09 \pm 0.16$ | $0.52 \pm 0.05$ | $0.71 \pm 0.05$ | $4.31 \pm 0.12$ | $0.59 \pm 0.05$ | $0.74 \pm 0.05$ |
| GPS | $6.62 \pm 0.01$ | $2.04 \pm 0.01$ | $2.61 \pm 0.00$ | $7.58 \pm 0.01$ | $3.14 \pm 0.01$ | $2.91 \pm 0.01$ |

Table 10: Performance of SCIGAN and the benchmarks when we increase the number of treatments in the dataset to 6 and 9. Bold indicates the method with the best performance for each dataset.

## M.5  Sample efficiency

We have also performed a further experiment to evaluate model performance in terms of sample efficency. For the MIMIC dataset, in Table 1 we report evaluation metrics for training SCIGAN with different number of training samples $N$ and evaluating on the same test set.

| | $\sqrt{\text{MISE}}$ | $\sqrt{\text{DPE}}$ | $\sqrt{\text{PE}}$ |
|---|---|---|---|
| $N = 100$ | $31.12 \pm 63.39$ | $7.72 \pm 2.57$ | $18.94 \pm 29.07$ |
| $N = 500$ | $13.36 \pm 10.46$ | $4.07 \pm 1.92$ | $2.63 \pm 0.94$ |
| $N = 1000$ | $3.80 \pm 1.04$ | $2.46 \pm 1.75$ | $1.03 \pm 1.13$ |
| $N = 1500$ | $2.95 \pm 0.37$ | $0.70 \pm 0.17$ | $0.63 \pm 0.12$ |
| $N = 1920$ | $2.09 \pm 0.12$ | $0.51 \pm 0.05$ | $0.32 \pm 0.05$ |

Table 11: Sample efficency analysis for MIMIC. Metrics are reported as Mean $\pm$ Std.

## M.6 Discrete dosage set-up and comparison with GANITE

In this set-up, we use the TCGA dataset and treatments 2 and 3 from Table 1 with dose-response curves $f_2(\mathbf{x}, d)$ and $f_3(\mathbf{x}, d)$ respectively. Let $\beta$ be the number of discrete dosages for which we want to generate data. We chose $\beta$ equally spaced points in the interval $[0, 1]$ as our set of discrete dosages: $\Delta = \{\frac{k}{\beta-1}\}_{k=0}^{\beta-1}$. To create factual dosages for our dataset, we sample the dosages as before $d_w \mid x \sim \text{Beta}(\alpha, \beta_w)$ (as described in Appendix J), and choose the closest discrete dosage from the set $\Delta$.

To evaluate SCIGAN in this setting, we maintain the same architecture for the multi-task generator and hierarchical discriminator. The only difference is that we now randomly sample dosages for the SCIGAN discriminator from $\Delta$.

We adopt the GANITE implementation proposed by [6]. To be able to have a fair comparison with SCIGAN we also use a multi-task architecture for the GANITE generator and we give as input to each multitask head the dosage parameter. The GANITE generator will generate outcomes for all possible discrete dosages in $\Delta$ and these will be passed to the GANITE discriminator to distinguish the factual one. For the GANITE generator we use a similar architecture to the SCIGAN generator with 2 hidden layers for each multitask head and 64 neurons in each layer. The GANITE discriminator consists of 2 fully connected layers with 64 neurons in each. We also set $\lambda = 1$. In addition, to maintain a similar set-up to SCIGAN, we train an inference network to learn the counterfactual outcomes with data from the GANITE generator. The inference network has the same architecture as the GANITE generator.

(a) $\sqrt{\text{DPE}}$        (b) $\sqrt{\text{PE}}$

Figure 10: Comparison between SCIGAN and GANITE in the discrete dosage set-up.

We clearly see from Fig. 10 that SCIGAN achieves a similar performance to GANITE for a small number of dosages ($< 6$) but then significantly outperforms GANITE for more dosages than 6. In fact, we see that while GANITE's performance degrades with an increasing number of dosages, SCIGAN's improves and then stabilises at around 12 dosages. This is due to the fact that the single discriminator in GANITE simply cannot handle a large number of dosages. Our hierarchical model, however, can. The worse performance of SCIGAN for the lower dosages can be attributed to the fact that for such few dosages (e.g. 3 dosages corresponds to only 6 total different interventions), the SCIGAN architecture is overly complex, and the sub-sampling of dosages for the discriminator is not actually necessary.

## M.7 Mixing dosage and no-dosage treatment options

We also evaluate the case when one of the treatments does not have a dosage parameter. For this experiment we generate data for the treatment that has a dosage parameter $d$ using $f_3(\mathbf{x}, d)$ and for the treatment without an associated dosage using $2C(\mathbf{v}_0^T \mathbf{x})$, where $\mathbf{v}_0$ are parameters, $\mathbf{x}$ are patient features and $C =$ is the scaling parameter. This set-up also corresponds to the scenario where we want to compare giving a treatment with a dosage and not giving any treatment.

SCIGAN can be easily extended to incorporate an additional treatment that does not come with a dosage parameter. Such treatments will not need a dosage discriminator but will be passed to the treatment discriminator. A head can be added to the generator for each such non-dosage treatment but will not need to take dosage as an input.

As the DRNet public implementation does not allow for this set-up, we compared SCIGAN with the multilayer perceptron model with multitask heads (MLP-M). This model is trained using supervised learning to minimize error on the factual outcomes and consists of two multitask heads: one head for the treatment option which receives as input the dosage and estimates the dose-response curve and one head for the no-treatment option.

As can be seen in Table 12, SCIGAN is capable of handling this setting and lends itself naturally to potentially mixed dosage and no-dosage treatment options.

| Method | TCGA | | | News | | | MIMIC | | |
|---|---|---|---|---|---|---|---|---|---|
| | $\sqrt{\text{MISE}}$ | $\sqrt{\text{DPE}}$ | $\sqrt{\text{PE}}$ | $\sqrt{\text{MISE}}$ | $\sqrt{\text{DPE}}$ | $\sqrt{\text{PE}}$ | $\sqrt{\text{MISE}}$ | $\sqrt{\text{DPE}}$ | $\sqrt{\text{PE}}$ |
| SCIGAN | **1.28** ± 0.09 | **1.37** ± 0.07 | **1.56** ± 0.06 | **3.18** ± 0.15 | **2.04** ± 0.09 | **2.49** ± 0.12 | **0.61** ± 0.08 | **1.82** ± 0.02 | **1.89** ± 0.03 |
| MLP-M | 2.08 ± 0.12 | 1.85 ± 0.16 | 2.02 ± 0.07 | 4.68 ± 0.11 | 2.45 ± 0.08 | 2.64 ± 0.08 | 1.56 ± 0.08 | 2.04 ± 0.03 | 2.14 ± 0.5 |

Table 12: Performance of individualized treatment-dose response when mixing treatment with no-treatment options. Bold indicates the method with the best performance for each dataset.

## M.8 Additional results on selection bias

In Fig. 11 we report the DPE for our treatment and dosage bias experiment from Section 6.5 of the main paper.

(a) Treatment selection bias  (b) Dosage selection bias

Figure 11: Additional performance metrics of the 4 methods on datasets with varying bias levels on TCGA dataset