[Reviews · NeurIPS 2020]

Review 1

Summary and Contributions: The paper proposes GAN-based framework which learns the distribution of the unobserved counterfactuals by estimating the effect of continuous interventions. The presented generator-discriminator framework effectively deal with estimating outcomes of continuous interventions. The paper also introduces a new semi-synthetic data simulation under the continuous intervention setting and validate their model on the dataset with pre-existing benchmarks.

Strengths: This paper is a well-written paper. It also conveys the main idea clearly. I found the paper clearly well written and very well presented. I love the idea of training the counterfactual generator and discriminator adversarially such that the generator competes with the discriminator by generating counterfactuals. They theoretically justified the presented solutions.

Weaknesses: More detailed description about the simi-synthetic dataset (e.g., information about feature, interventions, and treatment) should be included in the appendix section for clinical clarity.

Correctness: The technical solutions presented in the paper look reasonable.

Clarity: The paper is well-written.

Relation to Prior Work: The paper clearly discuss the connections to the pre-existing work by addressing their limitations and how they tackle the existing problems behind them.

Reproducibility: Yes

Additional Feedback: I really enjoy reading this paper. I find the idea of simultaneously estimating counterfactual outcomes for continuous interventions very interesting and I believe their proposed methodology can have a significant impact on the domains of healthcare: estimating an individual-level response to dosage can be very practical and useful in clinical decision-making situations. [Additional comment] Authors answered most of my raised concerns. I would like to have this paper accepted.


Review 2

Summary and Contributions: The paper addresses the important problem of estimating the effect of continuous treatment variables, provides theoretical guarantees for the proposed solution, and significantly improves over state-of-art baselines. The writing is clear and the experiment section is exhaustive.

Strengths: [S.N] Novelty: existing work for estimating treatment effects is mostly concerned with binary/categorical treatments whereas the proposed method can handle continuous as well as discrete interventions. [S.E] Empirical evaluation: thorough. [S.R] Relevance: important for adaptation of machine/deep learning in sensitive domains such as medicine.

Weaknesses: [W. N] Novelty: seemes like the paper draws a lot of inspiration from related works ([2, 6]), however the contributions of the SCIGAN are clear to me wrt empirical and theoretical results. [W. E. 1] Empirical: in the conclusion (line 308) the authors mention, “SIGAN needs a few thousands of training samples”, I was wondering if there is a sample efficiency experiment for this result? [W. E. 2] Appendix K: Was PCA used for all baselines or just for GPS, and is there a difference in the results with/without PCA? [W. E. 3] Will there be a version of the code, results, simulated data and generators be available?

Correctness: To the best of my knowledge, yes.

Clarity: Yes, only few minor comments, see "Additional feedback".

Relation to Prior Work: Yes this has been made clear throughout the text and also with additional results in the Appendix.

Reproducibility: Yes

Additional Feedback: I wonder if the authors considered the issues of callibration wrt the data distibuiton, can this have an impact on the results for estimating ITE? It has been shown that the generator distribution does not match the true data distribution (for example [1], [2]), which can be accounted for with custumized models. [1] Dai, Zihang, et al. "Calibrating energy-based generative adversarial networks." arXiv preprint arXiv:1702.01691 (2017). [2] Hitaj, Briland, et al. "Passgan: A deep learning approach for password guessing." International Conference on Applied Cryptography and Network Security. Springer, Cham, 2019. Minor: - Maybe replace "Intervention" with "Treatment" in Fig 1. - There is no bold annotation in Table 10. - I'm not sure I understand why use the term "hierarchical" and not "ensamble" of discriminator networks. - Maybe rename Theorem 2 in Appendix (currently enumerated same as in the main text). - Could you elaborate more on your motivation to include the permutation invariance and equivariance for the discriminator?


Review 3

Summary and Contributions: The current work proposes SCIGAN to learn the distribution of the unobserved counterfactuals. While previous works are focused on estimating the effect of discrete interventions, this work focuses on the continuous-valued intervention setting.

Strengths: 1. This work leverages the generative adversarial networks to estimate the effect of continuous-valued intervention, which can also be applied to the discrete case. 2. A hierarchical discriminator is proposed to handle the complexity of multiple-treatment setting and the structure of the continuous intervention setting. 3. It has been shown theoretically that the learned counterfactuals agree with the true data in marginal distribution. 4. Experimental results show superior performance compare to other methods.

Weaknesses: 1. One of the issues with the current presentation of the paper is that notations are very messy and hard to follow. It would have been better if a simplified notations were used. 2. Many of the details are omitted from the paper and left to appendices. 3. The stopping criteria for the GAN network (e.g. IS score, FID distance etc.) is not discussed. 4. Did authors face any of the famous common issues with training GANs such as gradient vanishing, mode collapse etc.? and if so, how did they avoid these issues for a stable training? 5. In the first paragraph of section 6, authors have mentioned that meaningful evaluation on real-world datasets is not feasible. If that’s the case, then what is the use of the current work in real applications? And isn’t that in contradiction with the author’s claim in the Introduction section about applicability of the work in many domains such as medical etc.?

Correctness: Yes

Clarity: Yes

Relation to Prior Work: Yes

Reproducibility: Yes

Additional Feedback:

[Author Response · NeurIPS 2020]

We would like to thank all reviewers for their valuable feedback which has helped us improve the paper!

**Reviewer 1:** ■ **Dataset descriptions:** Please note that Appendix I contains details about the datasets used in terms of
patient features, possible interpretations of treatments and of patient outcomes. We also provide links to the publicly
available datasets. Upon acceptance, we will release the code for the model and for the semi-synthetic data generation.

**Reviewer 2:** ■ **Novelty:** Our method only draws inspiration from [6] in terms of using a GAN framework to learn
counterfactual outcomes. Nevertheless, to handle continuous interventions, we propose a novel hierarchical discriminator
architecture. We also provide theoretical results, which are lacking from [6] to show that the proposed GAN framework
can indeed learn the distribution of the counterfactual outcomes. Finally, we introduce a new semi-synthetic data
simulation that can be used to benchmark causal inference methods for estimating the effects of continuous interventions.

■ **Sample efficiency:** Our remark regarding sample efficiency was
perhaps a bit offhand. Experimentally, we showed that SCIGAN works
for a few thousand samples (using the MIMIC dataset with 1920 training
samples). We had not investigated how SCIGAN performs below this
number. We have now performed a further experiment to evaluate model
performance in terms of sample efficency. For the MIMIC dataset, in
Table 1 we report evaluation metrics for training SCIGAN with different
number of training samples $N$ and evaluating on the same test set.

|  | $\sqrt{\text{MISE}}$ | $\sqrt{\text{DPE}}$ | $\sqrt{\text{PE}}$ |
|---|---|---|---|
| $N = 100$ | $31.12 \pm 63.39$ | $7.72 \pm 2.57$ | $18.94 \pm 29.07$ |
| $N = 500$ | $13.36 \pm 10.46$ | $4.07 \pm 1.92$ | $2.63 \pm 0.94$ |
| $N = 1000$ | $3.80 \pm 1.04$ | $2.46 \pm 1.75$ | $1.03 \pm 1.13$ |
| $N = 1500$ | $2.95 \pm 0.37$ | $0.70 \pm 0.17$ | $0.63 \pm 0.12$ |
| $N = 1920$ | $2.09 \pm 0.12$ | $0.51 \pm 0.05$ | $0.32 \pm 0.05$ |

Table 1: Sample efficency analysis for MIMIC.
Metrics are reported as Mean $\pm$ Std.

■ **PCA for GPS model:** PCA is only used for the GPS model for TCGA and News datasets, which contain a large
number of features, to reduce computational complexity. Since GPS is a linear method, using PCA as a pre-processing
step helps avoid problems with co-linear features. We used a publicly available implementation for GPS based on
the causaldrf package in R. After re-running GPS without PCA on News we obtained similar results to the ones in
Table 3 in the paper: $6.03 \pm 0.01$ ($\sqrt{\text{MISE}}$), $6.83 \pm 0.01$ ($\sqrt{\text{DPE}}$) and $22.56 \pm 0.03$ ($\sqrt{\text{PE}}$). ■ **Code and data:** We
will release the code for the model and semi-synthetic data generation upon acceptance. ■ **Calibration:** We did not
consider calibration - though it would certainly be an interesting future research direction. We would note, though,
that [R1] is not about improving the generator but rather gleaning a useful discriminator from the training procedure
(which would normally result in a degenerate discriminator), which could be used at test-time to evaluate the generated
response-curves. We will add discussion about this in the conclusion. ■ **Hierarchical discriminator:** The term
hierarchical refers to the fact that there are 2 levels to our discrimination procedure - (1) determine the factual treatment;
(2) determine the factual dosage given the factual treatment. In contrast with the term ensemble which would typically
refer to several models performing the same task, we have different models performing different tasks. ■ **Permutation**
**invariance and equivariance:** We use permutation invariance and equivariance because we are fundamentally dealing
with dose-response curves, which are themselves functions. To treat these as functions, we treat them as sets of points
of the form (input, output). For this reason we use permutation invariance and equivariance - so that the networks act as
functions on sets (rather than functions on vectors which would be the case without the in-/equi-variance).

**Reviewer 3:** ■ **Presentation and notation:** Please note that the problem we are aiming to solve requires complex
notation due to the fact that we are handling treatments with continuous dosage. Moreover, our choice of architecture in
terms of the hierarchical discriminator also needs complex notation. Unfortunately, we do not feel that the notation
can be simplified much. Appendix B contains a table for all of our notation and we will work further to improve the
presentation and notation in the revised manuscript. ■ **Details in appendices:** Due to the page limit for the conference,
it was not possible to add all of the details in the main paper. We have tried to keep as much information as possible in
the main paper, which involved many tough decisions about what was best placed in the main paper and what could
be placed in the appendix. ■ **Stopping criteria for the GAN network:** For all experiments with SCIGAN, we used
5000 training iterations for the GAN network. This number of training iterations was chosen to ensure convergence
of the generator loss, discriminator loss, as well as of the supervised loss. We will include details about the number
of training iterations used in the paper. ■ **Issues with GAN training:** We have not encountered gradient vanishing
problems when training our SCIGAN. It is not clear to us how the problem of mode collapse would even present itself
in this setting as we are not discriminating between entirely real and entirely fake samples. ■ **Evaluation on real data**
**and real-world applicability:** In real datasets, we only observe the outcome for the patient for a specific setting of the
treatment and the dosage. The counterfactual outcomes, i.e. the patient outcomes under different possible interventions,
cannot be observed. This is why it is not possible to use real data to evaluate how well the methods can estimate the
entire dose-response curve for each patient. However, this does not mean that this method cannot be deployed in real
world environments. In this regard, the problem is no different to the very well studied problem of treatment effect
estimation for a binary/categorical treatment [6, 16, R2] for which there is a *wealth* of existing literature containing
many examples of real-world applications. Evaluation on semi-synthetic data is standard for causal inference methods.

[R1] Dai, Zihang, et al. "Calibrating energy-based generative adversarial networks." preprint arXiv:1702.01691 (2017).
[R2] Jennifer L. Hill, "Bayesian nonparametric modeling for causal inference." JCGS, 2011


[Meta-Review · NeurIPS 2020]

The paper studies the problem of estimating the effect of continuous treatment variables. The authors propose a GAN-based framework to learns the distribution of the unobserved counterfactuals. The reviewers found the theoretical contribution as well as the simulation showing improvement over the pre-existing benchmarks satisfying. Estimating the effect of a treatment is a central problem to causal inference and as such this paper could be of interest to the broader NeurIPS audience.